https://doi.org/10.1038/s41467-021-21940-8　　**OPEN**

# Activation of GPR37 in macrophages confers protection against infection-induced sepsis and pain-like behaviour in mice

Sangsu Bang [1,5], Christopher R. Donnelly[1,5], Xin Luo[1,5], Maria Toro-Moreno [2,5], Xueshu Tao[1], Zilong Wang [1], Sharat Chandra[1], Andrey V. Bortsov[1], Emily R. Derbyshire [2] & Ru-Rong Ji [1,3,4 ✉]

GPR37 was discovered more than two decades ago, but its biological functions remain poorly understood. Here we report a protective role of GPR37 in multiple models of infection and sepsis. Mice lacking *Gpr37* exhibited increased death and/or hypothermia following challenge by lipopolysaccharide (LPS), Listeria bacteria, and the mouse malaria parasite *Plasmodium berghei*. Sepsis induced by LPS and Listeria in wild-type mice is protected by artesunate (ARU) and neuroprotectin D1 (NPD1), but the protective actions of these agents are lost in *Gpr37*$^{-/-}$ mice. Notably, we found that ARU binds to GPR37 in macrophages and promotes phagocytosis and clearance of pathogens. Moreover, ablation of macrophages potentiated infection, sepsis, and their sequelae, whereas adoptive transfer of NPD1- or ARU-primed macrophages reduced infection, sepsis, and pain-like behaviors. Our findings reveal physiological actions of ARU in host cells by activating macrophages and suggest that GPR37 agonists may help to treat sepsis, bacterial infections, and malaria.

[1] Center for Translational Pain Medicine, Department of Anesthesiology, Duke University Medical Center, Durham, NC, USA. [2] Department of Chemistry, Duke University, Durham, NC, USA. [3] Department of Neurobiology, Duke University Medical Center, Durham, NC, USA. [4] Department of Cell Biology, Duke University Medical Center, Durham, NC, USA. [5]These authors contributed equally: Sangsu Bang, Christopher R. Donnelly, Xin Luo, Maria Toro-Moreno. ✉email: ru-rong.ji@duke.edu

Inflammation occurs as a result of tissue damage or localized infection, leading to an inflammatory cascade of reactions ultimately aimed at restoring organismal homeostasis. Precise regulation of this process is critical, as under- or over-activation of the immune response can lead to damaging consequences. For example, failure to mount an effective immune reaction against a localized infection can lead to inadequate pathogen control, resulting in systemic inflammation, and generating the sequelae, which ultimately lead to sepsis and death[1]. Seemingly paradoxically, sepsis is driven chiefly as a result of acute hyper-activation of the immune system, producing a systemic cytokine storm that can produce severe consequences, including fever, hypothermia, multiple organ failure, and death[2]. Of note, sepsis is remarkably challenging to treat, with mortality rates of approximately 20% in high-income countries, a number which is likely much larger in poor-resource areas. Individuals who do survive sepsis often do so with long-lasting cognitive and physical impairments[3]. Infection also causes pain through direct or indirect activation of nociceptor neurons by bacteria[4,5]. Thus, there is a critical need to identify strategies to combat infection-induced sepsis and its sequelae.

A growing body of evidence indicates that resolution of inflammation is an active process involving the production of lipid-derived specialized pro-resolving mediators (SPMs), which include resolvins, protectins, and maresins[6]. The resolution of inflammation is emerging as a therapeutic strategy for inflammation-coupled diseases[7–9], including pain[10]. Additionally, SPMs have recently been shown to augment control of infections in animal models. For example, Resolvin D2 (RvD2) is a potent regulator of leukocyte activation and exhibited efficacy in controlling sepsis[11]. In addition, SPMs have been shown to be dynamically regulated following bacterial challenge in mice and reduce antibiotic requirements[12]. SPMs have also been shown to augment host defense against viruses, as protectin D1/neuroprotectin D1 (PD1/NPD1) was demonstrated to protect against lethal influenza virus infection[13]. SPMs have also been proposed as a potential therapeutic strategy to complement the clinical management of severe coronavirus disease (COVID-19) caused by SARS-CoV2, as a key mechanism in the pathophysiology of COVID-19 appears to be pulmonary hyper-inflammation leading to life-threatening cytokine storms[14].

To exert their biological actions, SPMs activate specific GPCRs to promote the resolution of inflammation[6]. GPR37, also known as parkin-associated endothelin-like receptor (Pael-R), was cloned in 1997 and is highly expressed in the brain and implicated in neurological disorders such as Parkinson's disease and autism[15,16]. However, the role of GPR37 in other cell types remains unclear. In a recent study, we found that GPR37 is expressed by macrophages (MΦ). Moreover, we identified GPR37 as a receptor for NPD1 and found that activation of GPR37 with NPD1 promotes MΦ phagocytosis and resolution of inflammatory pain[17]. However, it remains unknown whether GPR37 also plays a role in regulating protective immunity against pathogens, infection, or sepsis.

In this study, we explored the role of GPR37 in sepsis using several mouse models, including high dose administration of systemic lipopolysaccharide (LPS, also known as endotoxin), bacterial infection with *Listeria monocytogenes* (*L.m.*), and infection with the mouse malaria parasite *Plasmodium berghei* (*P.b.*). In all three models, we found that host GPR37 confers protection against infection, sepsis, and sepsis-associated adverse events, including serum cytokine production, hypothermia, pain, and death. We also found that the 1st-line anti-malaria drug, artesunate (ARU), can bind GPR37, and activation of GPR37 with NPD1 or ARU decreases infection severity and septic death. Additionally, we demonstrate that these protective effects are primarily mediated by GPR37 in macrophages, and activation of GPR37 promotes macrophage phagocytosis and clearance of pathogens. Macrophage ablation substantially impaired survival following challenge with *P.b.*, whereas adoptive transfer of NPD1- or ARU-primed *Gpr37*[+/+] macrophages improves survival. Thus, our study suggests that GPR37 agonists may be a prospective therapeutic strategy to treat sepsis, infections, and malaria. Our findings also reveal a surprising host-intrinsic role of ARU in promoting protective immunity through GPR37 in macrophages.

## Results

**NPD1 and GPR37 are protective in mouse models of sepsis.** We first examined whether protectin D1/neuroprotectin-D1 (PD1/NPD1) could regulate sepsis in wild-type (WT) and *Gpr37*[−/−] mice following LPS and *L.m.* infection. To avoid confusion of PD1 abbreviation with programmed death protein 1 (PD-1), we used NPD1 for protectin D1 in this study. In WT mice, intraperitoneal (I.P.) injection of 10 mg/kg LPS caused mortality in 94% mice at 7 days (Fig. 1a, b). This mortality was reduced by pretreatment of NPD1 (500 ng I.P.; $P = 0.0043$, Fig. 1b). Notably, this protective effect of NPD1 was completely abolished in *Gpr37*[−/−] mice ($P = 0.0080$, Fig. 1b). As expected, LPS also resulted in sepsis-induced hypothermia in WT mice 24 h after administration, but this hypothermia was significantly protected by NPD1 in WT mice ($P = 0.0019$, Fig. 1c). Notably, NPD1 did not prevent hypothermia in KO mice ($P = 0.0002$, Fig. 1c). We also examined serum levels of the pro-inflammatory cytokine IL-6, which is regarded as both a marker and a mediator contributing to the pathophysiology of sepsis[18]. LPS resulted in a 1000-fold increase in serum levels of IL-6, and this increase was significantly reduced by NPD1 in WT mice ($P = 0.0022$, Fig. 1d), but not in *Gpr37*[−/−] mice (Fig. 1d).

To substantiate these findings using a more physiological model of sepsis, we challenged mice with I.P. injection of $1 \times 10^7$ CFUs of *Listeria monocytogenes* (*L.m.*, Fig. 1e), a gram-positive bacterium that causes Listeriosis in mammalian hosts. We found that WT mice also exhibited high mortality (90%) 7 days after *L.m.* infection, which was protected by NPD1 ($P = 0.0396$, Fig. 1f). This protection was abolished in *Gpr37*[−/−] mice (WT + NPD1 vs *Gpr37*[−/−] + NPD1, $P < 0.0001$, *Gpr37*[−/−] vs *Gpr37*[−/−] + NPD1, $P = 0.5525$, Fig. 1f). Notably, we also observed a significant increase in survival in vehicle-treated WT mice compared to vehicle-treated *Gpr37*[−/−] mice ($P = 0.0217$), suggesting GPR37 may confer protection in the absence of exogenously administered NPD1 (Fig. 1f), possibly due to the presence of endogenous GPR37 activators. *L.m.* infection also resulted in hypothermia in WT mice 6 h following inoculation, and this effect was significantly protected by NPD1 treatment in WT mice ($P = 0.0433$, Fig. 1g). The effect of NPD1 in KO mice was completely abolished ($P < 0.0001$, Fig. 1g). *L.m.* also resulted in a 500-fold increase in serum IL-6 levels, which was attenuated by NPD1 treatment in WT mice ($P = 0.0255$, Fig. 1h). This attenuation was abolished in KO mice ($P < 0.0001$, Fig. 1h). Collectively, these data indicate that NPD1 confers protection against infection and septic death in a GPR37-dependent manner.

In a previous study, we demonstrated that NPD1 activates GPR37 in macrophages[17]. To determine whether macrophages confer the protective effects of NPD1 against sepsis in vivo, we treated peritoneal macrophages (pMΦ) from WT and *Gpr37*-KO mice with NPD1 (30 nM) and measured pro-inflammatory cytokines in response to challenge with vehicle, LPS (1 μM), or *L.m.* ($1 \times 10^4$ CFUs). We observed both LPS and *L.m.* dramatically increased secretion of TNF-α, IL-1β, and IL-6 in culture media, and these increases were inhibited by NPD1 treatment in WT but not in KO macrophages (Fig. 1i–k). These results suggest that NPD1 may act on macrophages to inhibit the systemic cytokine storm during sepsis.

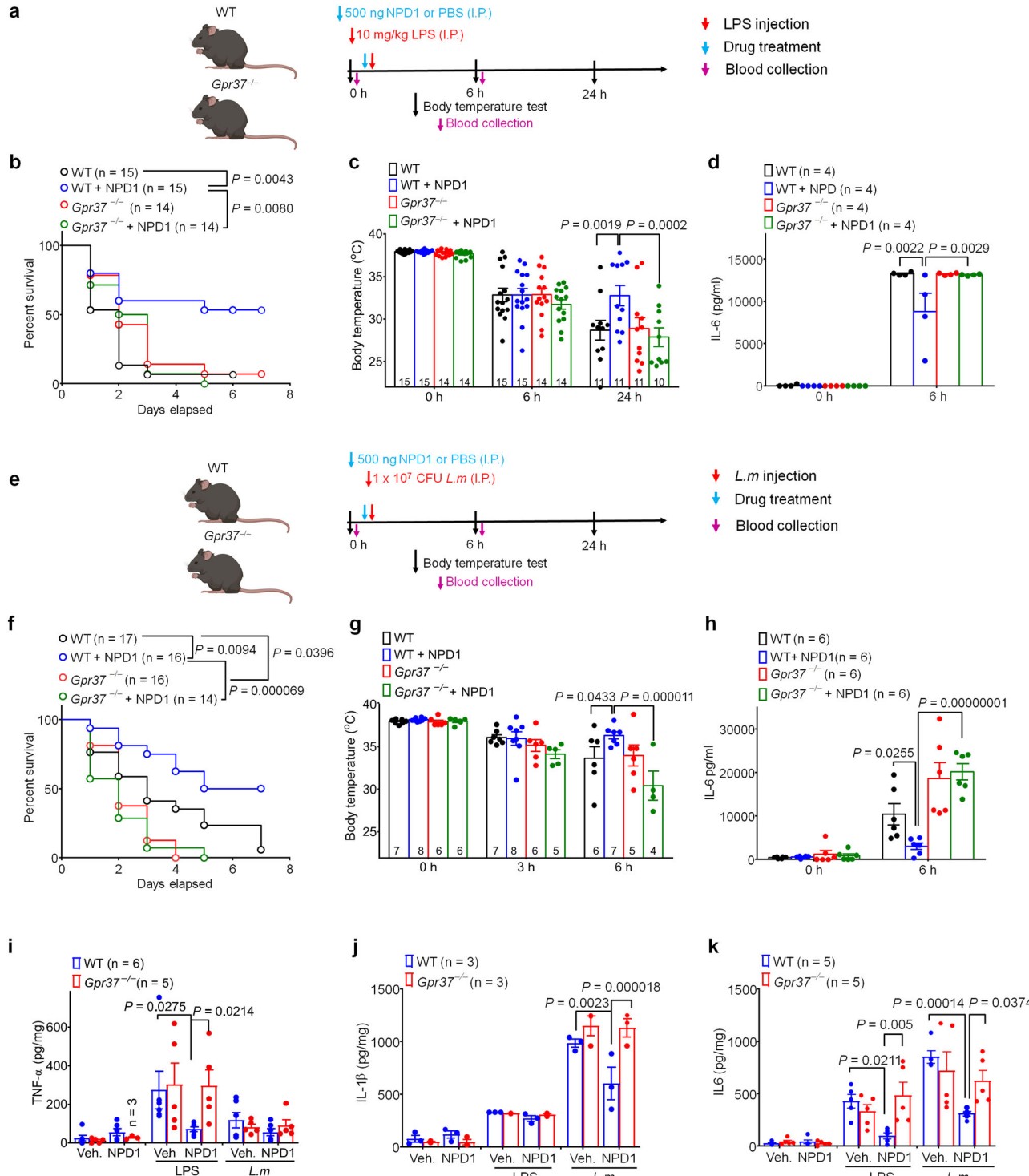

**Fig. 1 Protection of LPS- and *L. monocytogenes*-induced sepsis by GPR37. a** Experimental design for the LPS model. Wild-type (WT) mice and *Gpr37*−/− mice were given intraperitoneal (I.P.) injection of LPS (10 mg/kg) together with vehicle or NPD1 (500 ng, I.P.) treatment. **b** Survival curves of WT and *Gpr37*−/− mice treated with vehicle or NPD1. Sample sizes are indicated in brackets. **c** Time course of rectal temperature in WT and *Gpr37*−/− mice. Sample sizes are indicated in columns. **d** Serum IL-6 levels in WT and *Gpr37*−/− mice ($n = 4$/group). **e** Experimental design for *Listeria monocytogenes* (*L. m*, $10^7$, I.P) inoculation in WT and *Gpr37*−/− mice, together with treatment of vehicle or NPD1 (500 ng, I.P.). **f** Survival curves of WT and *Gpr37*−/− mice treated with NPD1 or Vehicle. Sample sizes are indicated in brackets. **g** Time course of rectal temperature in WT and *Gpr37*−/− mice with vehicle or NPD1 treatment. **h** Serum IL-6 levels in WT and *Gpr37*−/− mice 6 h after *L. m* injection ($n = 6$). **i–k** Cytokine levels in culture media of pMΦ from WT or *Gpr37* mice. pMΦ cultures were treated with LPS (1 μM, 37 °C, 12 h) or *L. m* ($1 \times 10^4$ CFU, 37 °C, 12 h) together with vehicle (Veh.) or 30 nM NPD1 (30 nM). ELISA was tested for TNF-α (**i**), IL-1β (**j**), and IL-6. **k** Data are expressed as mean ± s.e.m. and statistically analyzed by Mantel–Cox test (**b, f**), Two-way ANOVA followed by Tukey's posthoc test (**c, d, h**) Bonferroni's post hoc test (**i–k**).

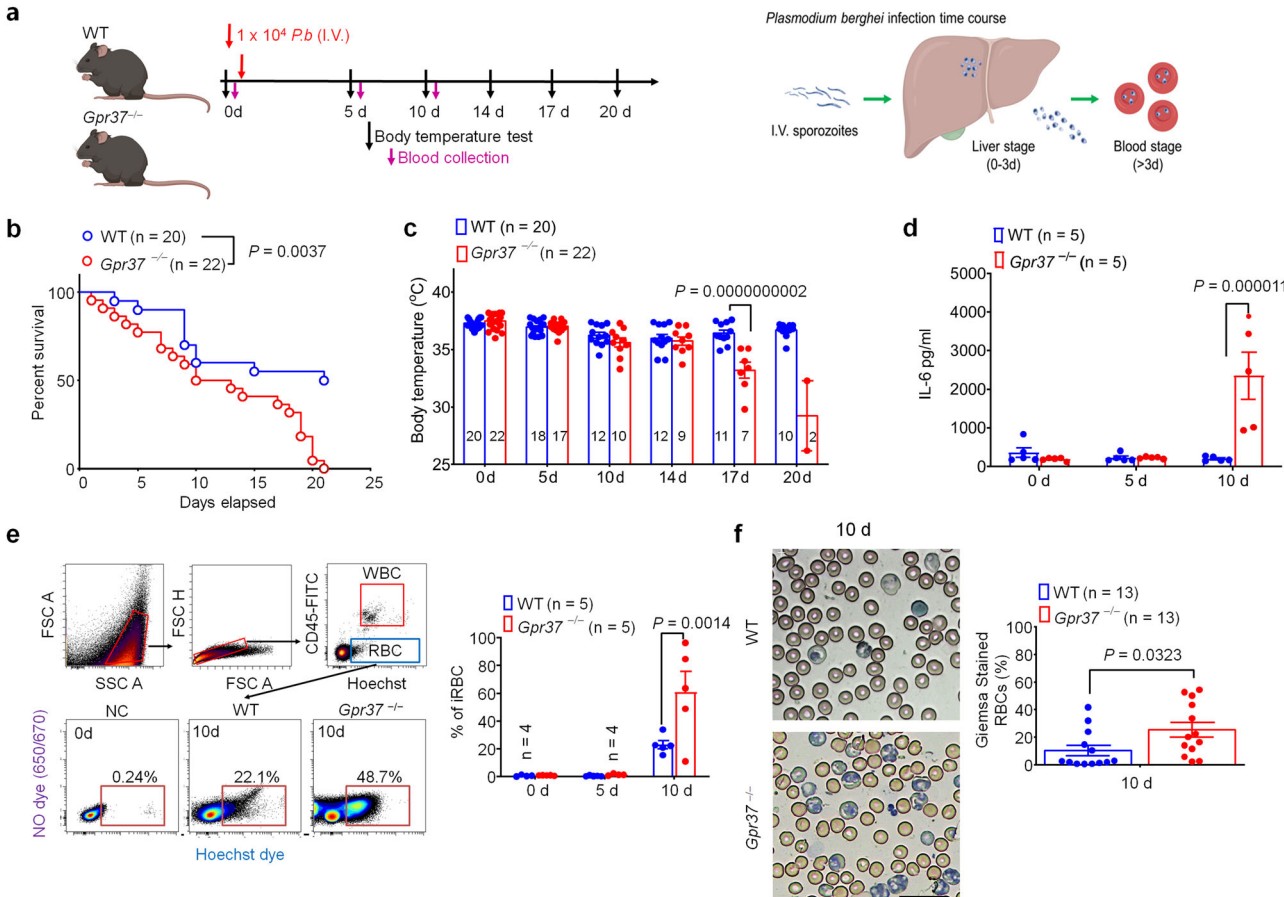

**Fig. 2 GPR37 confers protection against malaria infection. a** Experimental design for the mouse model of malaria infection. WT and *Gpr37*⁻/⁻ mice were treated with *Plasmodium berghei* (*P.b.*) sporozoites (10⁴, I.V.) and survival, rectal temperature, and serum IL-6 were measured as outcomes of infection. **b** Survival curves of WT and *Gpr37*⁻/⁻ mice. Sample sizes are indicated in brackets. **c** Time course of rectal temperature in WT and *Gpr37*⁻/⁻ mice. **d** Serum IL-6 levels in WT and *Gpr37*⁻/⁻ mice (*n* = 5/group). **e** Flow cytometry analysis of peripheral blood samples collected from WT and *Gpr37*⁻/⁻ mice 10 d after *P.b.* infection. Infected RBCs (iRBCs) were incubated with Hoechst dye (1 µg/ml) and CD45-FITC. Left upper panel: gating strategy to remove debris, aggregated cells, CD45⁺ WBCs, and small particles (<3 µm). Left down panel: representative images of RBCs sorted from WT or *Gpr37*⁻/⁻ mice 10 d after *P.b.* inoculation. Right panel: quantification of flow cytometry analysis at the indicated timepoints showing the proportion of iRBCs (Hoechst⁺ RBCs/total RBCs; 1 × 10⁶ RBC analyzed per sample, *n* = 5). **f** Giemsa staining was performed to detected iRBCs in WT and *Gpr37*⁻/⁻ mice 10 d after *P.b* infection. Left panel shows representative images. Right panel shows quantification of Giemsa-stained iRBCs (*n* = 13/group). Scale bar, 20 µm. Data are expressed as mean ± s.e.m. and statistically analyzed by Mantel–Cox test (**b**), 2-way ANOVA followed by Bonferroni's post hoc test (**c–e**), and unpaired two-tailed *t*-test (**f**).

**GPR37 is protective against malaria infection.** Malaria, an infectious disease caused by protozoan parasites of the *Plasmodium* genus, remains a substantial cause of death in many areas of the world according to the 2019 World malaria report (https://www.who.int/malaria/publications/world-malaria-report-2019/en/). To test whether GPR37 confers protection against malaria infection, we inoculated WT or *Gpr37*⁻/⁻ mice with 1 × 10⁴ sporozoites of *Plasmodium berghei* (*P.b*) via intravenous (I.V.) injection (Fig. 2a). Notably, this *Plasmodium* species only infects mice and presents with a disease course that is similar to the *Plasmodium falciparum* parasite, which causes severe malaria infections in humans[19]. Parasite inoculation caused ~50% mortality in WT mice at 10 days, with little subsequent change in the mortality rate by 21 days (Fig. 2b). In contrast, *P.b.*-infected *Gpr37*⁻/⁻ mice exhibited similar mortality at 10 days, but much higher mortality at 21 days (*P* = 0.0037, vs. WT mice), reaching an overall survival rate of 0% (Fig. 2b). Compared to the LPS and *listeria* models, the body temperature change was mild after *P.b.* infection, and *Gpr37*⁻/⁻ mice exhibited significant hypothermia at later stages in the disease (day 17, *P* < 0.0001, vs. WT mice, Fig. 2c). We also observed significantly increased weight loss in KO mice 14 days post-infection

(*P* = 0.0048, vs. WT; Supplementary Fig. 1a). In addition, *P.b.*-infected *Gpr37*⁻/⁻ mice exhibited significantly higher serum IL-6 levels 10 days post-inoculation (*P* < 0.0001, vs. WT mice, Fig. 2d), although serum levels of TNF-α and IL-1β were not changed (Supplementary Fig. 1b, c). Importantly, no differences were observed between WT and *Gpr37*⁻/⁻ mice in the proportion of macrophages, B-cells, T-cells, or neutrophils under steady-state conditions (Supplementary Fig. 1d–f).

Malaria-causing *Plasmodium* parasites undergo a stereotypical course of disease originating with a liver stage followed by a bloodborne stage, which is characterized by infection of red blood cells (RBC), leading to a loss of RBCs and anemia (Fig. 2a, right schematic)[20]. We measured the progression of *P.b.* sporozoite infection to the blood stage using multiple parameters (Fig. 2b–f), including flow cytometry (Fig. 2e) and Giemsa staining (Fig. 2f). Infected RBCs were detected by incubation with Hoechst dye, which stains *P.b.*-infected RBCs, as RBCs lack nuclei. RBCs were also characterized as CD45⁻ cells isolated from peripheral blood. Flow cytometry analysis of whole blood from WT mice revealed an RBC infection rate of 23% 10 days after *P.b.* sporozoite inoculation, but this rate was significantly higher in KO mice (61%, *P* = 0.0014,

Fig. 2e). Giemsa staining further confirmed an increase in the RBC infection ratio in KO mice ($P = 0.0323$, vs. WT, Fig. 2f). Collectively, these results suggest that GPR37 may protect against malaria infection and its downstream consequences.

**Artesunate interacts with GPR37 and enhances macrophage phagocytosis.** To identify additional putative GPR37 agonists, we conducted $Ca^{2+}$ imaging, cAMP- BRET, and pERK-FRET assays in GPR37-expressing HEK293 cells, using a small library of 138 natural compounds (10 μM, Supplementary Fig. 2a, Supplementary Table 1). All three assays revealed artesunate (ARU), an artemisinin (ART) derivative, as a potential ligand (Supplementary Fig. 2a, b). Notably, ARU exerts direct antiparasitic actions against *Plasmodium* parasites and is a 1st-line drug for the treatment of severe malaria and, in combination with mefloquine, is an artemisinin-based combination therapy recommended by the WHO for uncomplicated *P. falciparum* malaria treatment[21]. GPR37-dependent activity was observed for ARU, but not ART or the known active ART-metabolite dihydroartemisinic acid (DHA) (Supplementary Fig. 2b). Based on these findings, we performed overlay assays to identify candidate ligands for GPR37, wherein putative binding partners were immobilized on a nitrocellulose membrane and incubated with purified human GPR37 (Supplementary Fig. 2c). Honokiol, ART, and ARU were tested, along with DMSO as a negative control. Interestingly, we observed strong and dose-dependent binding of ARU to GPR37 (Supplementary Fig. 2c).

Next, we conducted computational modeling of the possible interaction between ARU and GPR37. Homology modeling of GPR37, a characteristic GPCR, was performed using the human M2-muscarinic acetylcholine and human delta opioid receptors crystal structures as templates. Molecular dynamics simulations suggest that ARU interacts with GPR37 at Q535, R401, and R418 to form hydrogen bonds (Fig. 3a–c). A similar analysis suggests NPD1 forms hydrogen bonds with R418, Y432, E508, and Q535 residues on GPR37 (Fig. 3d–f). Notably, ARU and NPD1 share the same binding sites of R418 and Q535 on GPR37 (Fig. 3a–f). The prosaptide Tx14 is a known peptide agonist of GPR37[17]. Interestingly, we were unable to confirm binding of Tx14 in the NPD1 binding site of GPR37, prompting us to search for other putative binding sites. We found the second extracellular loop (ECL2) to be a putative binding site, and additional docking experiments indicated that Tx14 is positioned at the distal side of ECL2 and Tx14 interacts with GPR37 at Q402 and Y339 to form hydrogen bonds (Fig. 3g–i). Overall, our binding prediction studies suggest that GPR37 hydrogen bond interaction with Q535 and R418 stabilize ARU and NPD1 binding and these specific interactions in the ligand binding site can discriminate between ARU analogs such as ART and dihydroartemisinin (DHA). Furthermore, Tx14 may interact with GPR37 via distinct binding sites.

Previously we demonstrated that activation of GPR37 by NPD1 promotes the resolution of zymosan-induced inflammation by augmenting MΦ phagocytosis[17]. We reasoned that, if ARU also acts as a ligand of GPR37, it should exert similar GPR37-dependent effects in macrophages. To this end, we investigated whether ARU could enhance phagocytosis of peripheral MΦ (pMΦ) using fluorescently-labeled zymosan particles. We observed that ARU treatment (30 μM for 30 min) increased uptake of zymosan particles by pMΦ ($P = 0.0005$, Fig. 4a, b), and this effect was abolished in pMΦ from $Gpr37^{-/-}$ mice ($P = 0.0008$, Fig. 4a, b). Additionally, we completed this experiment instead using *L.m.* incubated with a pH-sensitive pHrodo dye to test whether ARU can promote pMΦ-mediated clearance of live bacteria ($1 \times 10^5$ CFUs). Similarly, we observed that ARU promoted pMΦ phagocytosis of *L.m.*, and this

effect was abolished in pMΦ cultured from $Gpr37^{-/-}$ mice ($P < 0.0001$, Fig. 4c, d). To determine whether ARU or NPD1 alter bacterial clearance in vivo, we employed several complementary measures. First, we measured bacterial load in peritoneal fluid (PF) from WT or $Gpr37^{-/-}$ mice 24 h after infection with *L.m.* We observed that both NPD1 and ARU substantially reduced bacterial load in PF, and this effect was abolished in $Gpr37^{-/-}$ mice (NPD1: $P = 0.0301$; ARU: $P = 0.0003$ for ARU; Fig. 4e, f).

Given that *L.m.* can evade killing and proliferate internally within macrophages and other phagocytes, we sought to determine whether activation of GPR37 with NPD1 or ARU could enhance elimination of *L.m.* by pMΦ. To this end, pMΦ from WT or $Gpr37^{-/-}$ mice were co-cultured with *L.m.* and treated with vehicle, NPD1, or ARU, and colony-forming unit assays were performed on pMΦ lysate. Interestingly, NPD1 and ARU treatment significantly reduced internal *L.m.* levels. Notably, these effects were abolished in pMΦ from $Gpr37^{-/-}$ mice, which also exhibited markedly greater internal *L.m.* compared to pMΦ from WT mice (Fig. 4g). We also measured bacterial load and PMN infiltration in PF, liver, and spleen in vehicle, NPD1, or ARU-treated mice at various timepoints after *L.m.* infection (Supplementary Fig. 3a). While only NPD1 significantly reduced bacterial load in PF, both NPD1 and ARU reduced tissue bacterial load in liver and spleen with varying kinetics (Supplementary Fig. 3b–d).

We also measured PMN infiltration as a readout of *L.m.*-induced inflammation and tested the pro-resolution actions of NPD1 and ARU in PF, liver, and spleen tissue. We observed that *L.m.* inoculation dramatically increased PMN levels in these cells and tissues of vehicle-treated mice, with peak PMN infiltration occurring at different timepoints for each tissue. Importantly, NPD1 and ARU treatment attenuated *L.m.*-induced PMN infiltration (Supplementary Fig. 3e–g). Using the Gaussian equation, we calculated $\psi_{max}$ (maximal PMNs), T-$\psi_{max}$ (time point when PMNs reach $\psi_{max}$), $\psi 50$ (50% of maximal PMNs), and T-$\psi_{50}$ (time point corresponding to 50% PMN reduction), followed by calculation of the resolution interval (the time between T-$\psi_{max}$ and T-$\psi_{50}$). We found that the resolution interval (Ri) was not significantly different between vehicle and GPR37 agonist-treated groups (vehicle: 11.07 h; NPD1: 18.00 h; ARU: 16.13 h; $P = 0.961$ vehicle vs. NPD1, $P = 0.9371$ vehicle vs. ARU for P.F; Supplementary Fig. 3h), in part due to a substantial inhibition of infection by NPD1 and ARU, making it difficult to see the peak response in the treated mice.

Given that ARU is a 1st-line antimalarial drug, and our finding that ARU can promote GPR37-dependent macrophage phagocytosis of zymosan particles and live *L.m.* pathogens, we posited that ARU may also promote MΦ-mediated clearance of *P.b.* Macrophages recognize malaria-infected RBCs through danger-associated molecular patterns (DAMPs) and engulf them by phagocytosis, subsequently clearing them via autophagosome formation[22–25]. To test whether ARU contributes to clearance of *P.b.*-infected RBCs, we examined two markers, microtubule-associated protein 1A/1B-light chain 3 (LC3), an autophagy/autophagosome marker, and nitric oxide (NO), a marker for reactive oxygen species. Importantly, both LC3 induction and NO production are critical for MΦ phagocytosis. Flow cytometry results showed that 30 μM ARU increased LC3 and NO expression in WT pMΦ but these effects were abolished in $Gpr37$ deficient pMΦ (Fig. 4h–k, Supplementary Fig. 4a). Of note, we also tested the effects of ARU in the THP1 human macrophage cell line. Following ARU treatment, differentiated THP1 cells exhibited increased engulfment of human RBCs infected with *P. falciparum* (*P.f.*) and exhibited heightened LC3 and NO activity ($P < 0.0001$ and $P = 0.0164$, respectively, Supplementary Fig. 4b, c).

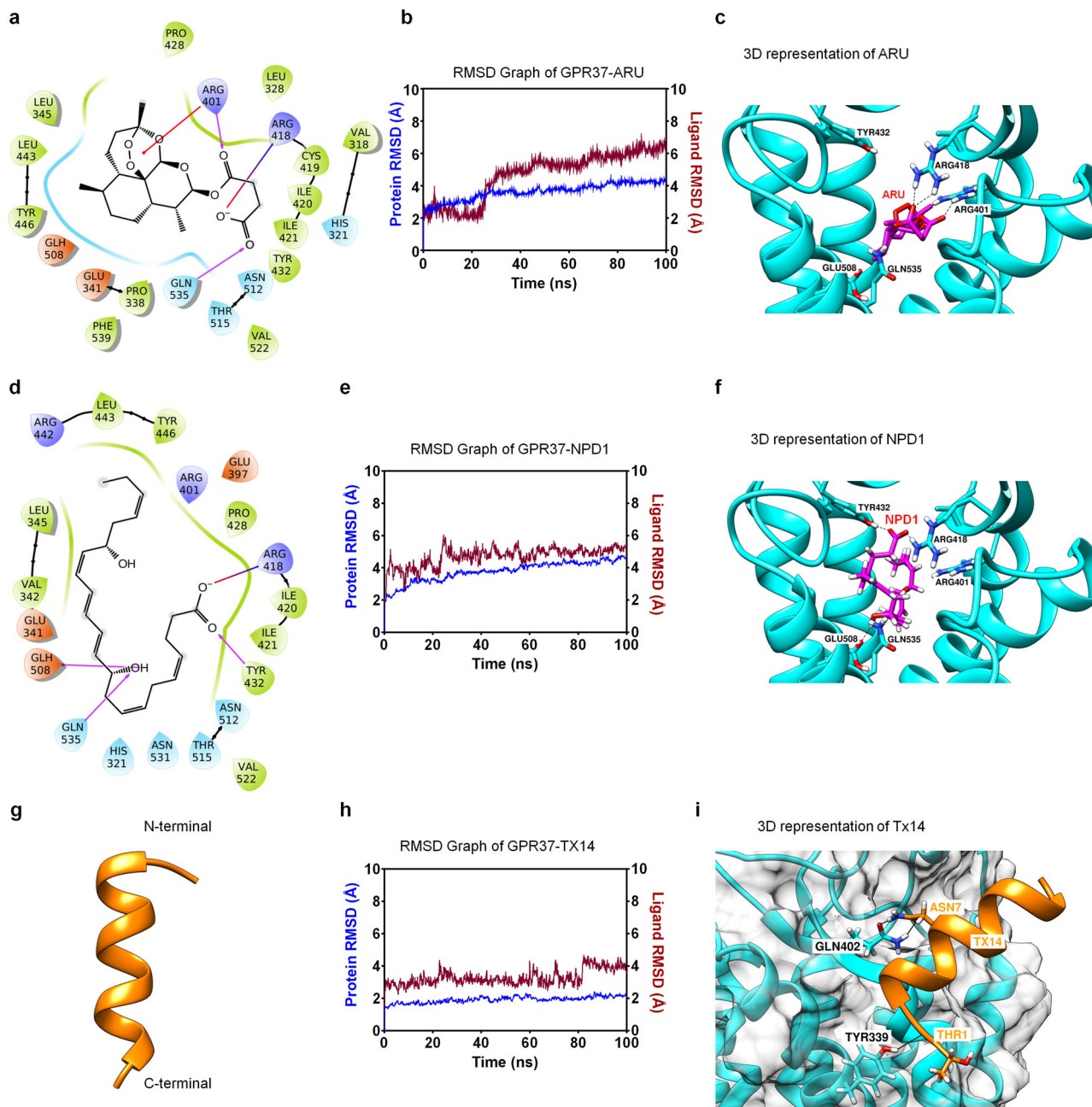

**Fig. 3 Computational simulation predicts distinct docking sites for each GPR37 ligand.** The docking modes showing potential binding sites at GPR37 for neuroprotectin D1 (NPD1, **a–c**), artesunate (ARU, **d–f**), and Tx14 (**g–i**). **a**, **d** 2D interaction map for ARU (**a**) and NPD1 (**d**) docked in the GPR37 model binding site. **b**, **e**, **h** Results of molecular dynamics simulation (MDS) for ARU (**b**), NPD1 (**e**), and Tx14 (**h**). Y-axis shows  root mean square deviation (RMSD graph) of the protein–ligand complex for 100 nanoseconds time. During the MDS, the dynamic stability of the GPR37–ARU complex was calculated using RMSD adjustments. The RMSD value of the protein backbone is stabilized at 4 Å after 23 ns and the RMSD of ARU jumped to 5 Å at around 25 ns and then showed a steady increase up to 7 Å until the end of the simulation. The protein backbone of the GPR37 was observed an RMSD of 4 Å during 80% trajectory time of the simulation, indicating that GPR37 remained stable. The RMSD value of the NPD1 is stabilized between 4 and 6 Å after 25 ns. **c**, **f**, **i** Three-dimensional (3D) representation of ARU (**c**), NPD1 (**f**), and Tx14 (**i**). **g** Modeled structure of Tx14 (orange). The top-scoring docked conformations of compounds NPD1 and ARU in the active site of the homology model of GPR37 were analyzed in terms of key residues involved in the interaction and their preferred mode of binding. All resulting docked complexes were further analyzed in detail. The NPD1 forms hydrogen bonding with R418, Y432, E508, and Q535 residues in the GPR37 whereas ARU forms a hydrogen bond with R401, R418, and Q535. The Tx14 forms hydrogen bonding with Q402 and Y339.

**Artesunate protects against sepsis via GPR37.** Our data indicate that activation of GPR37 confers protection in multiple models of sepsis, and ARU can bind GPR37 and induce GPR37-dependent phagocytosis in vitro. Based on these findings, we next sought to determine whether ARU has protective effects in the LPS and *L.*

*m.* models of sepsis in the absence of malaria. In LPS-induced sepsis (Fig. 5a), ARU-treated WT mice had significantly improved survival compared to ARU-treated KO mice ($P = 0.0003$, Fig. 5b). LPS-induced hypothermia was also alleviated by ARU in WT mice ($P = 0.0002$, Fig. 5c) but the ARU effect was

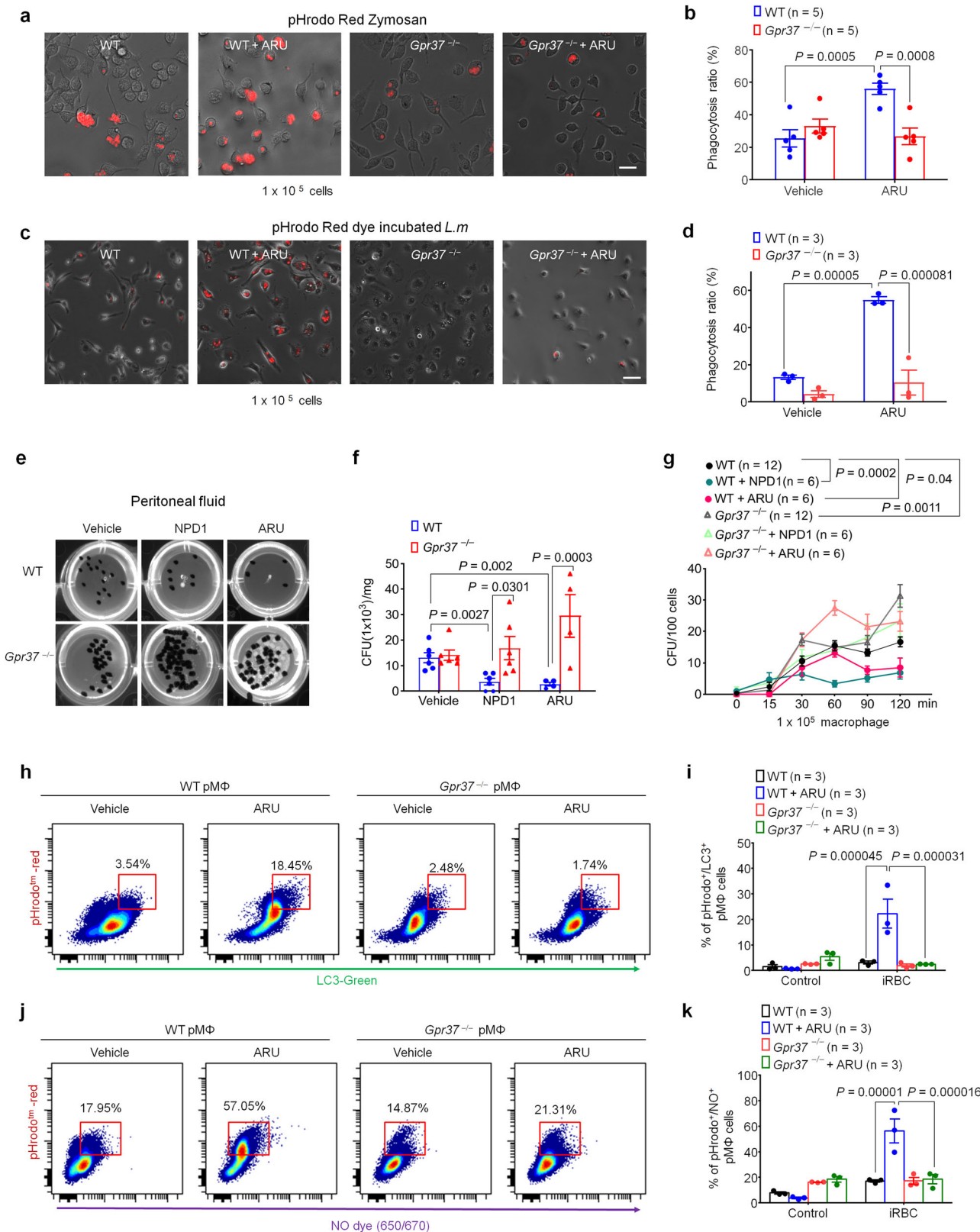

abolished in KO mice 24 h post-inoculation ($P < 0.0001$, Fig. 5c). Additionally, compared with ARU-treated WT mice, ARU-treated KO mice also had significantly higher IL-6 levels ($P = 0.0009$, Fig. 5d). We next tested the effects of ARU in the *L.m.*-induced model of sepsis (Fig. 5e). Notably, ARU also improved survival in WT mice ($P = 0.0145$, Fig. 5f) but the effect of ARU

was abolished in *Gpr37*-KO mice ($P = 0.0311$, Fig. 5f). ARU also protected against *L.m.*-induced hypothermia ($P = 0.0194$, Fig. 5g) and IL-6 production ($P = 0.0072$, Fig. 5h) in WT mice. The protective effect of ARU against hypothermia was abolished in KO mice ($P = 0.0007$, Fig. 5g), as in the LPS model. Together, these data suggest that ARU confers GPR37-dependent protection in

**Fig. 4 ARU promotes macrophage phagocytosis via GPR37. a** Representative images of in vitro phagocytosis assay in which pHrodo zymosan particles ($1 \times 10^5$) were incubated with pMΦ from WT or $Gpr37^{-/-}$ mice treated with vehicle or ARU (30 μM, 30 min, 37 °C). Red fluorescence indicates intracellular update indicating phagocytosis. Scale bars: 10 μm. **b** Quantification of pMΦ phagocytic activity of zymosan positives cells ($n = 5$/group). **c** Representative images of an in vitro phagocytosis assay in which fluorescence-labeled live ($1 \times 10^5$ CFU) $L.m.$ were incubated with $1 \times 10^4$ pMΦ from WT or $Gpr37^{-/-}$ mice treated with vehicle or ARU, as in (**a**). Scale bars: 10 μm. **d** Quantification of pMΦ phagocytic activity of fluorescently-labeled live $L.m.$ ($n = 3$/group). **e** Representative image of in vivo bacterial colony formation unit assay in which $L.m.$ ($1 \times 10^7$ CFU) were inoculated in WT or $Gpr37^{-/-}$ mice treated with vehicle, NPD1, or ARU (I.P.) at 24 h. **f** Quantification of CFU level in peritoneal fluid after $L.m.$ infection ($n = 6$ for vehicle or NPD1/group, $n = 4$ for ARU/group). **g** Time dependent internal bacterial load after $L.m.$ and pMΦ from WT or $Gpr37^{-/-}$ mice co-culture ($1 \times 10^5$ cells and $1 \times 10^4$ CFU $L.m.$, $n = 12$ for control, $n = 6$ for NPD1 or ARU). **h** pMΦ from WT or $Gpr37^{-/-}$ mice were treated with vehicle or ARU (30 μM, 1 h; 37 °C), followed by incubation with iRBCs labeled with pHrodo dye ($10^6$ cells, 1 h; 37 °C). Flow cytometry analysis of iRBC phagocytosis by pMΦ was performed by analyzing pHrodo in conjunction with the autophagosome marker LC3 sensitive dye (see gating strategy in Supplementary Fig. 4a). **i** Quantification of the proportion of LC3$^+$/pHrodo$^+$ pMΦ ($10^5$ cells analyzed; $n = 3$ mice/group). **j** Flow cytometry analysis was performed as in (**h**), instead using pHrodo$^+$ in conjunction with a NOS-sensitive dye to reveal phagocytosis. **k** Quantification of the proportion of NO$^+$/pHrodo$^+$ pMΦ ($1 \times 10^5$ cells analyzed; $n = 3$ mice/group (**k**). Data are expressed as mean ± s.e.m. and statistically analyzed by two-way ANOVA followed by Bonferroni's (**b**, **d**) or Tukey's (**f**, **g**, **i**, **k**) post-hoc test.

different models of sepsis, and thus, may be of therapeutic value beyond the treatment of malaria. Of note, ARU appears to have the most protective effects in the $L.m.$ infection model, compared to the LPS and $P.b.$ models.

To test whether acute administration of ARU at the onset of the disease could prevent malaria disease progression in our model, we inoculated mice ($1 \times 10^4$ $P.b.$ sporozoites, IV) on day 1 followed by treatment with ARU (10 mg/kg, IP, daily) on days 2–4 (Fig. 5i). This treatment paradigm did not reduce $P.b.$-induced death in WT mice or KO mice (Fig. 5j). Instead, regardless of ARU treatment, KO mice had worse survival than WT mice. ARU protected hypothermia in WT mice ($P = 0.0196$) but not in KO mice on post-infection day 14 (Fig. 5k). Serum IL-6 levels at 10 days post-infection showed a difference between untreated WT and KO mice ($P = 0.0128$), but not between ARU-treated WT and KO mice (Fig. 5l). ARU treatment also significantly decreased the percent of infected RBC (iRBC) in WT mice ($P = 0.0494$) but not in KO mice on day 10 (Supplementary Fig. 4d, e). Although neither treatment with NPD1 alone or combinatorial treatment with NPD1 and ARU significantly altered survival, the combination treatment led to a greater reduction in the proportion of iRBCs (Supplementary Fig. 4f–k). Collectively, this course of treatment with ARU exhibits a moderate protective effect in our malaria model, although it does not significantly improve the survival rate in WT mice, consistent with the previous reports[26].

**Macrophages are required for protection against malaria-induced sepsis.** To test whether macrophages themselves are required to protect against malaria infection, we depleted macrophages via I.P. injection of clodronate liposomes 2 days after $P.b.$ sporozoite administration (Fig. 6a), a timepoint chosen as sporozoites begin emerging from the liver between days 2 and 3. Mice treated with clodronate-liposomes exhibited significantly higher mortality following $P.b.$ infection compared with control liposome-treated mice ($P = 0.0061$, Fig. 6b). Clodronate treatment in $P.b.$-infected mice also resulted in increased hypothermia on day 10 ($P < 0.0001$, Fig. 6c) and caused a surge in IL-6 levels ($P = 0.0136$, Fig. 6d). Following clodronate ablation of macrophages, we also observed an increased percentage of infected RBCs by flow cytometry ($P = 0.0147$, Fig. 6e, f). Notably, we also performed flow cytometry to confirm that clodronate liposome administration successfully ablated circulating macrophages ($P = 0.0084$, Fig. 6g, h). These data indicate that the macrophages are required for protection against malaria infection. Moreover, given that clodronate treatment results in only a temporary depletion of macrophages[27], these data suggest that macrophages are particularly important for malaria parasite clearance early in the course of the infection.

**Adoptive transfer of ARU- or NPD1-treated macrophages alleviates sepsis and malaria.** To test whether GPR37 in macrophages mediates protection against malaria infection, we employed an adoptive transfer experiment by injecting $5 \times 10^4$ pMΦ (I.P.) into $Gpr37^{-/-}$ mice, 2 days after $P.b.$ sporozoite inoculation (Fig. 7a). Before the transfer, pMΦ were pre-treated (primed) with ARU (30 μM), NPD1 (30 nM), or vehicle (1% DMSO) for 24 h. Transplantation of $Gpr37$-deficient MΦ with ARU pretreatment did not improve the survival rate; all mice died by 18 days post-$P.b.$ infection (Fig. 7b). Notably, the median survival day increased from 13.5 d to 18 d after adoptive transfer of ARU-primed WT-pMΦ ($P = 0.0472$, vs. vehicle-treated WT pMΦ, Fig. 7b). The median survival day further increased from 13.5 d to 21 d following adoptive transfer of NPD1-primed WT-pMΦ (Fig. 7b). Concordantly, ARU and NPD1-primed WT pMΦ reduced the proportion of infected RBCs ($P = 0.0051$, $P = 0.0390$, respectively, vs ARU primed $Gpr37^{-/-}$ pMΦ, Fig. 7c, d). ARU or NPD1 primed pMΦ also protected against $P.b.$-induced IL-6 surge ($P = 0.0063$, $P = 0.0293$, respectively, Supplementary Fig. 5a). However, adoptive transfer of vehicle or ARU-primed $Gpr37$-deficient pMΦ to WT mice (Supplementary Fig. 5b) did not show significant protective effects on mortality ($P = 0.7684$, Supplementary Fig. 5c) and infection rate ($P > 0.05$, Supplementary Fig. 5d), arguing these effects are GPR37-dependent. Additionally, $L.m.$-induced septic death and sepsis-associated comorbidities (hypothermia, systemic IL-6 surge) were significantly attenuated by the transfer of NPD1-primed WT pMΦ in both WT and $Gpr37^{-/-}$ mice (Fig. 7e–g and Supplementary Fig. 6). However, $Gpr37$-lacking macrophages and non-primed macrophages failed to produce anti-septic effects in WT and $Gpr37^{-/-}$ mice (Supplementary Fig. 6). Altogether, these results demonstrate that activation of GPR37 in macrophages is critical for relieving $L.m.$ induced sepsis and $P.b.$ mediated death.

**GPR37-expressing macrophages reduce pain caused by bacterial infection.** A common comorbidity of bacterial infection is pain[28]. Since pain behaviors are typically experimentally measured in the hindpaw of rodents, we performed intraplantar injection of 10 μl $L.m.$ ($5 \times 10^6$ CFUs) in WT and $Gpr37^{-/-}$ mice (Fig. 8a), followed by behavioral testing to measure mechanical allodynia, thermal hyperalgesia, or cold allodynia (Fig. 8b). Hindpaw $L.m.$ infection in WT mice induced robust edema (Fig. 8c) and pain hypersensitivity, as indicated by mechanical allodynia in the von Frey test (Fig. 8d), cold allodynia in the plantar acetone test (Fig. 8e), and heat hyperalgesia in the Hargreaves test (Fig. 8f). Compared with WT mice, $Gpr37^{-/-}$ mice exhibited enhanced $L.m.$-induced pain behaviors in each of the tests (Fig. 8d–f), despite similar mechanical and cold

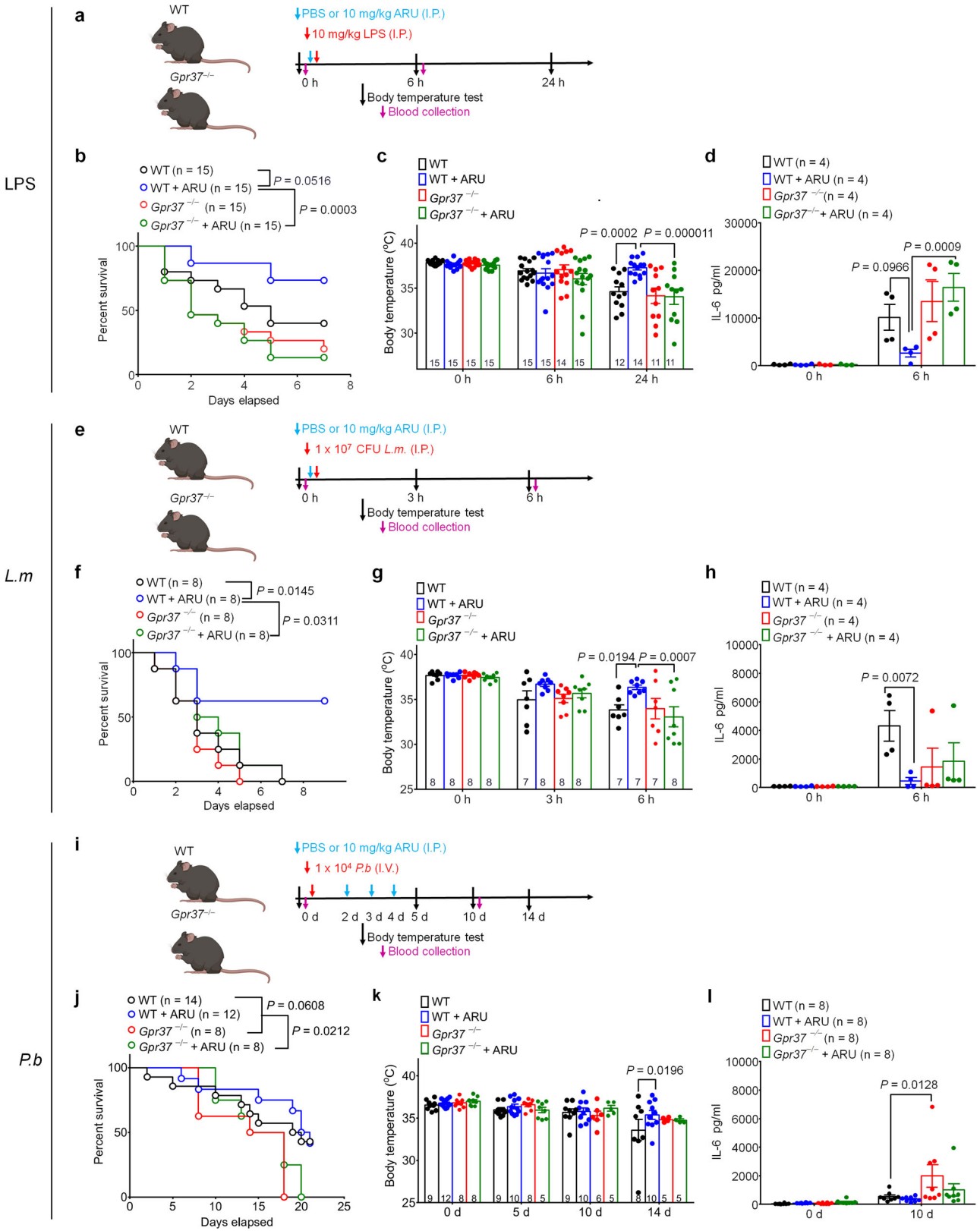

sensitivity at baselines. Remarkably, mechanical allodynia recovered by day 14 in WT mice, but remained unresolved in *Gpr37*<sup>−/−</sup> mice (Fig. 8d). Finally, the adoptive transfer of ARU-primed pMΦ also reversed *L.m.*-induced mechanical allodynia in *Gpr37*<sup>−/−</sup> mice (Fig. 8g, h), and intraplantar injection of ARU

(100 μg) partially reversed the infection-induced mechanical allodynia in WT but not *Gpr37*<sup>−/−</sup> mice (Fig. 8i). These results demonstrate that (1) GPR37 is also protective against pain caused by bacterial infection and (2) ARU can alleviate pain after infection via GPR37.

**Fig. 5 Artesunate protects mice from infection and septic death in a Gpr37-dependent manner. a** Experimental design to test ARU in the LPS model. WT and *Gpr37*[−/−] mice were administered LPS (10 mg/kg I.P.) together with vehicle or ARU (10 mg/kg). **b** Survival curves of WT and *Gpr37*[−/−] mice treated with vehicle or ARU. Sample sizes are indicated in brackets. **c** Time course of rectal temperature in WT and *Gpr37*[−/−] mice. Sample sizes are indicated in columns. **d** Serum IL-6 levels in vehicle and ARU-treated WT and *Gpr37*[−/−] mice (n = 4/group). **e** Experimental design to test ARU in the *Listeria monocytogenes* (*L.m.*, 10[7], I.P.) infection model in WT and *Gpr37*[−/−] mice. **f** Survival curves of WT and *Gpr37*[−/−] mice treated with vehicle or ARU. **g** Time course of rectal temperature in WT and *Gpr37*[−/−] mice in the *L.m.* infection mice. Sample sizes are indicated in columns. **h** Serum IL-6 levels in vehicle and ARU-treated WT and *Gpr37*[−/−] mice (n = 4/group). **i** Experimental design to test ARU in the *P.b.* (1 × 10[4], I.V.) infection model in WT and *Gpr37*[−/−] mice. **j** Survival curves of WT and *Gpr37*[−/−] mice treated with vehicle or ARU. **k** Time course of rectal temperature in WT and *Gpr37*[−/−] mice. Sample sizes are indicated in columns. **l** Serum IL-6 levels in vehicle and ARU-treated WT and *Gpr37*[−/−] mice (n = 8/group). Data are expressed as mean ± s.e.m. and statistically analyzed by Mantel–Cox test (**b, f, j**), Two-way ANOVA followed by Turkey's post-hoc test (**c, d, g, h, k, l**).

## Discussion

In our previous study, we demonstrated that GPR37 activation by NPD1 promotes macrophage phagocytosis and inflammatory pain resolution[17]. In this study, we extend these findings, demonstrating that GPR37 confers protection against sepsis or infection following challenge with LPS, bacterial infection by *L.m.*, or parasite infection by *P.b.* Critically, adoptive transfer of macrophages primed with GPR37 activators was sufficient to decrease signatures of infection and septic death, and these effects were abolished in mice lacking GPR37. Although at present no conditional knockout tools exist to confirm these findings, these results strongly suggest that GPR37 in macrophages is sufficient to confer protection against sepsis and/or infection in these models. Notably, while LPS and *L.m.* inoculation are well established models of sepsis, we are cautious in our use of this terminology in the *P.b.* model of malaria, as it is unclear whether this model follows a typical course of sepsis and septic death. Regardless, GPR37 and macrophages also confer protection against *P.b.* infection in this model, attenuating severity of infection, hypothermia, and death.

Malaria remains a public health threat, especially in Southeast Asia, where artemisinin-based combination therapies (ACTs) are losing their efficacy[29]. Although not fully understood, the antimalarial mechanism of action of artemisinin involves activation of the endoperoxide bridge by ferrous heme, which accumulates as the parasite degrades hemoglobin. This reactive species can subsequently alkylate parasite proteins and lipids to cause lethality[30]. Recent studies have shown that ARU has an anti-inflammatory role[31] in experimental cerebral malaria[32] and experimental colitis[33] as well as human rheumatoid arthritis[34]. However, these studies failed to demonstrate the molecular targets of ARU. Interestingly, it was proposed that in vivo artesunate treatment causes rapid decline in parasitemia by promoting phagocyte-mediated clearance of parasitized RBCs[35]. Critically, our results establish that GPR37 can also be activated by ARU. Importantly, ARU promotes phagocytosis in macrophages in vitro and robustly improves survival and decreases hallmarks of sepsis induced by LPS or *L.m.* in vivo. Thus, our data indicate ARU may also be of therapeutic benefit in sepsis. Notably, administration of ARU-primed macrophages was also sufficient to mitigate *P.b.* infection in a GPR37-dependent manner. While currently administered as a treatment measure due to its potent direct antiparasitic actions our data suggests ARU may also play a more complex role in in vivo malaria infections. Based on our results, this effect may be limited to ARU and not observed in other commonly used analogs such as ART.

The molecular mechanisms by which ARU interacts with GPR37 and the downstream signaling of GPR37 are still elusive. Even though GPR37 is regarded as an orphan GPCR and its structure is not yet fully resolved, the docking results showed the interaction between ARU and GPR37 in a specific binding pocket involving Q535 and R418 (Fig. 3), which is conserved in the transmembrane region of the closest relative of GPR37, GPR37L1[36]. Our docking experiment also shows the interaction of Q535 and R418 with NPD1 (Fig. 3b).

Interestingly, according to our structural prediction analyses, Tx14, a previously recognized peptide ligand of GPR37, may interact with GPR37 via distinct binding sites. Downstream signaling of GPR37 is likely to involve phosphorylation of ERK and AKT pathways and phagocytosis[17]. Further studies are warranted to investigate the signaling mechanisms by which NPD1 and ARU binding to GPR37 to produce their beneficial effects.

The risk of death from sepsis is as high as 30% affecting about 49 million people worldwide in 2017, with 11 million deaths[37]. According to the World Health Organization 2019 Report, malaria claims more than 400,000 lives every year (https://www.who.int/publications-detail/world-malaria-report-2019). In this study, we identified an immunotherapy mechanism for sepsis and malaria by which artesunate and NPD1 activate macrophages to promote phagocytosis via GPR37. Taken further, we posit that GPR37 agonists could be developed to treat infection and/or sepsis after bacterial and parasite infections as well as infection-induced pain (Fig. 9). Notably, SPMs have been shown to mitigate infection and sepsis in different animal models[6,11], and NPD1, in particular, was shown to be effective in preventing H5N1 influenza[13]. Therefore, GPR37-based immunotherapy may help to treat sepsis and virus infections. Interestingly, both SPMs and ARU and its derivatives have been suggested as potential therapeutics for COVID-19 following SARS-CoV2 infection[14,38]. Early emerging data suggests that ARU and/or its derivatives may inhibit SARS-CoV2 replication in vitro[39,40], leading to a small-scale clinical trial to evaluate the efficacy of ARU in patients with mild COVID-19 (NCT04387240). While studies thus far have suggested the actions of ARU may be due to direct effects on SARS-CoV2 virus replication, it will be of critical importance to evaluate whether ARU may also produce protective effects on COVID-19, especially in an in vivo context where the role of innate and adaptive immune cells can also be evaluated.

Last but not least, infection-induced pain is not well studied[41]. HIV infection is associated with chronic pain, peripheral neuropathy, and chronic neuroinflammation in the spinal cord[42], while other viruses, such as varicella zoster virus (VZV) and herpes simplex virus (HSV) potently infect primary sensory neurons and produce inflammatory lesions resulting in intense pain, which can be attenuated by resolvin E1[43]. SARS-CoV2 infection is not associated with acute pain but may result in chronic pain in COVID-19 patients due to high systemic stress and persistent neuroinflammation[44,45]. Notably, NPD1 was shown to potently reduce inflammatory pain and neuropathic pain in mice[46,47], and we also showed that ARU-priming of macrophages could promote pain resolution after post-infection, and ARU treatment itself could rapidly alleviate pain after bacterial infection through GPR37 (Fig. 8h, i). These experiments support the notion that ARU-mediated activation of GPR37 in macrophages promotes pain resolution, adding to the emerging breadth of literature supporting the important role of macrophages in a variety of pain conditions[48]. Moreover, given that ARU can suppress infection-induced pain within 1 h of treatment, it is also possible that ARU

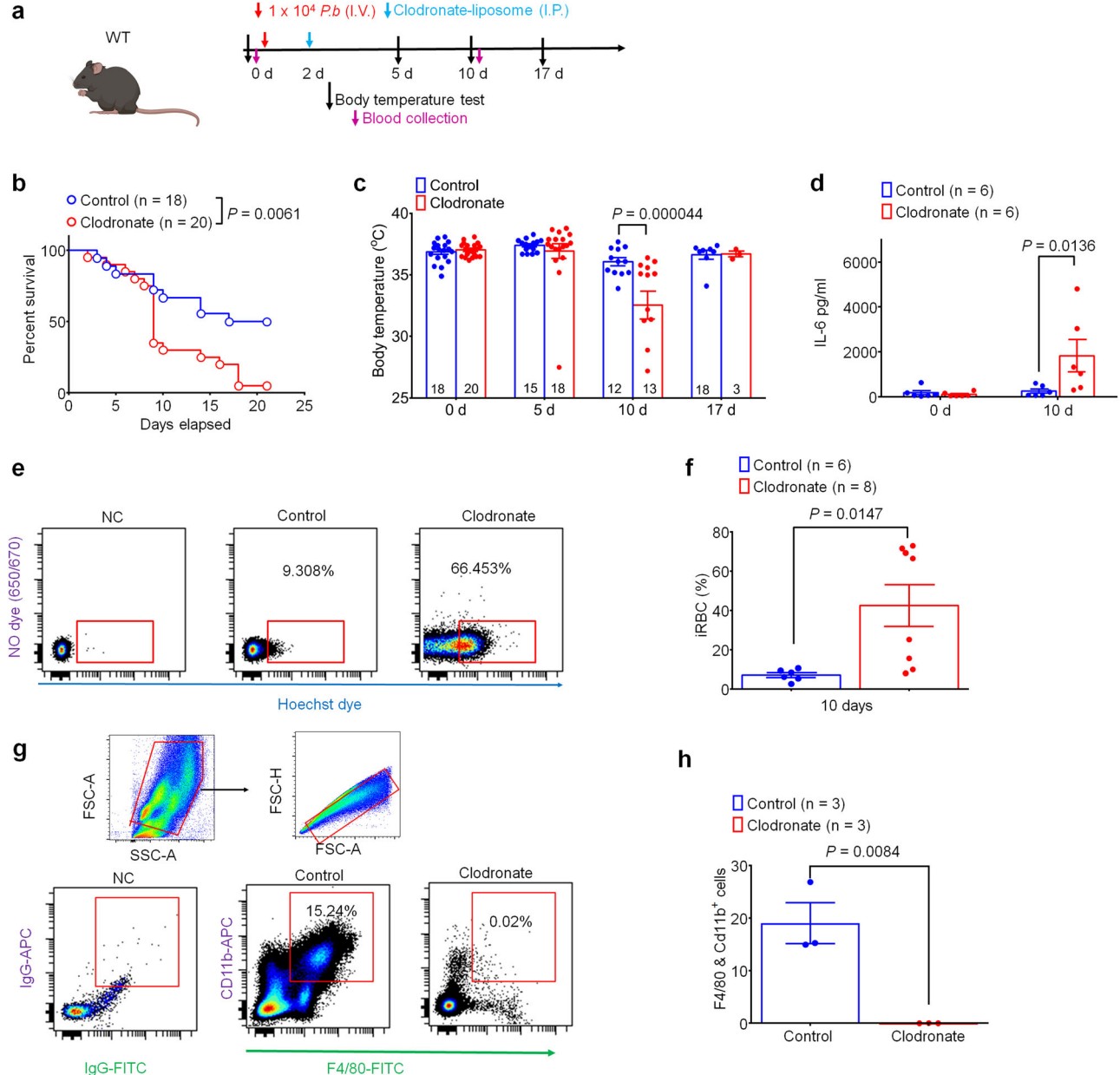

**Fig. 6 Macrophages confer protection in the *P.b.* mouse model of malaria. a** Experimental design to deplete macrophages following *P.b.* inoculation using clodronate liposomes (100 μl I.P.) in WT mice. **b** Survival curves of control liposome or clodronate liposome-treated mice. Sample sizes are indicated in brackets. **c** Time course of rectal temperature in control- or clodronate-liposome-treated mice. Sample sizes indicated in columns. **d** Serum IL-6 levels in control- or clodronate liposome-treated mice 10d after *P.b.* inoculation ($n = 6$/group). **e**, **f** Flow cytometry analysis to measure the proportion of *P.b.* infected RBCs (iRBCs) 10 d after inoculation, determined by positive staining following incubation with Hoechst dye (1 μg/ml) and negative staining for NO dye (650/670, 2 μl). **e** Representative images of flow cytometry results. (see gating strategy in Supplementary Fig. 4d). **f** Quantification of iRBCs (Hoechst+ RBCs/total RBC) 10 d after *P.b.* inoculation in control liposome- or clodronate liposome-treated mice (1 × 10^6 RBC analyzed per sample, sample sizes indicated in brackets). **g** Representative flow cytometry images of peritoneal macrophages (Cd11b+/F4/80+ population) 5 d after control liposome or clodronate liposome administration, as in (**a**) ($n = 3$ mice/group; 1 × 10^5 cells analyzed per sample) with gating strategy. **h** Quantification of peritoneal macrophages in control liposome- or clodronate liposome-treated mice. Data are expressed as mean ± s.e.m. and statistically analyzed by Mantel–Cox test (**b**), Two-way ANOVA followed by Bonferroni's post-hoc test (**c**, **d**), or unpaired two-tailed *t*-test (**f**, **h**).

directly inhibits pain through actions on GPR37 in peripheral sensory neurons, which is an intriguing future direction. Thus, we propose that NPD1 and other GPR37 agonists could be utilized to treat acute and chronic pain conditions post-infection as well as in other clinical contexts (e.g., neuropathic pain).

## Methods
**Reagents**. We obtained the following reagents from the indicated sources/vendors: pHrodo® Red Zymosan Bioparticles® Conjugate (Thermo Scientific; P35364, 1 mg/ml PBS stock), pHrodo® Red AM (Thermo Scientific; P35372, 10 mM DMSO stock), natural drug library (Selleckchem; L1400, 10 mM DMSO stock, see Supplementary Table 1 for drugs tested), LC3 detection dye (Dojindo Molecular

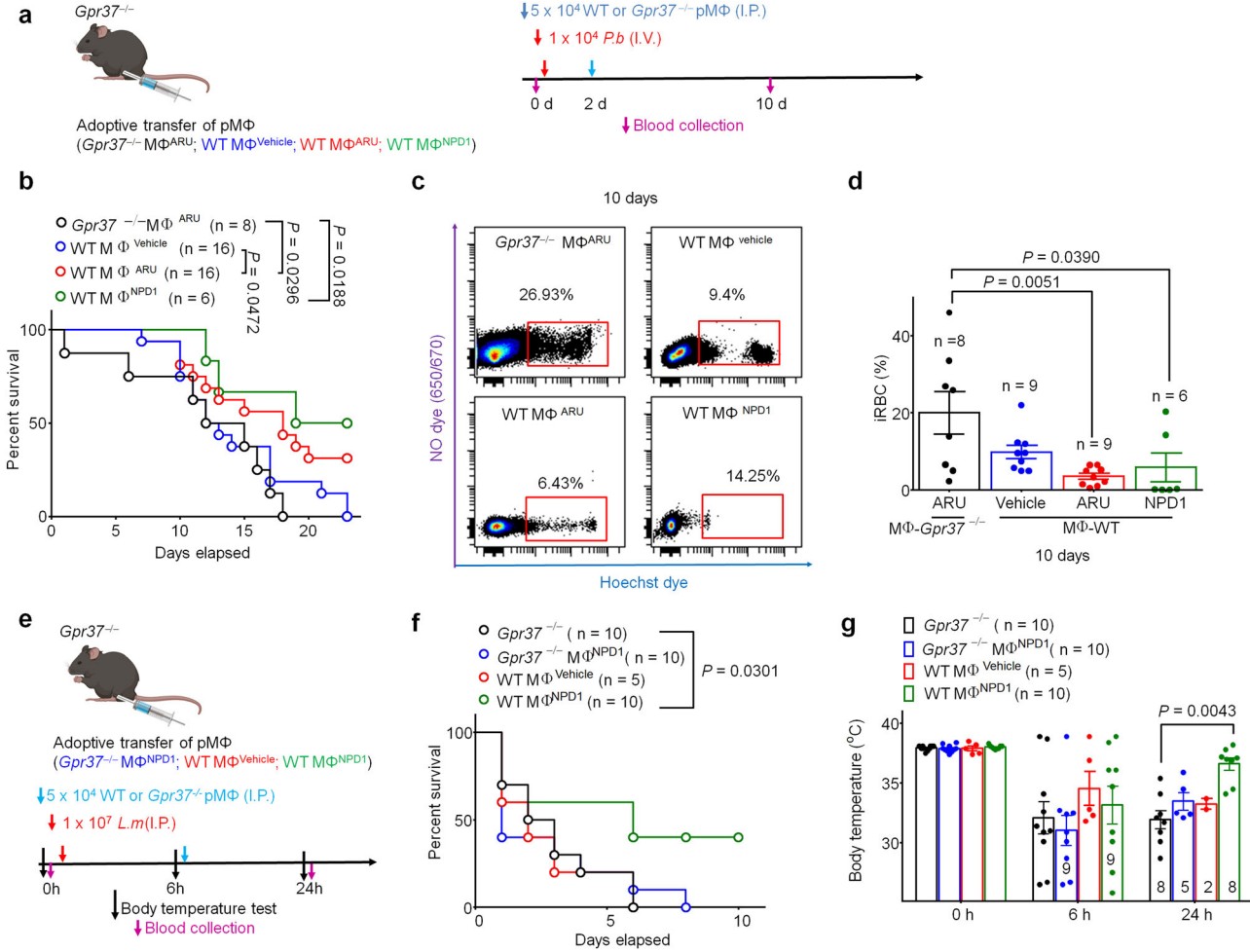

**Fig. 7 Adoptive transfer of macrophages primed with Gpr37 activators confers protection against malaria and listeria infection. a** Experimental design to test whether adoptive transfer of pMΦ primed with Gpr37 activators can attenuate malaria infection. WT or *Gpr37*−/− pMΦ were primed with vehicle, ARU (30 μM), or NPD1 (30 nM) for 24 h, followed by washout of the drug and adoptive transfer of pMΦ (5 × 10⁴, I.P.) into *P.b.*-inoculated *Gpr37*−/− mice (1 × 10⁴ sporozoites, I.V.) 2 d after infection. **b** Survival curves of *Gpr37*−/− mice treated with vehicle-, ARU-, or NPD1-primed WT pMΦ. As a control, *Gpr37*−/− mice were also treated with ARU-primed pMΦ from *Gpr37*−/− mice. Sample sizes are indicated in brackets. **c, d** Flow cytometry analysis of peripheral blood samples taken from the mice outlined in a 10 d after *P.b.* inoculation. Infected RBCs (iRBCs) were detected using Hoechst dye (1 μg/ml) and negative for NO dye (650/670). **c** Representative images of flow cytometry reveal the proportion of iRBCs (Hoechst⁺/NOS⁻) in each treatment group (see gating strategy in supplementary Fig. 4d). **d** Quantification of iRBCs in each group (10⁶ RBCs analyzed per sample), *n* = 6–9 mice/group as indicated. **e** Experimental design for adoptive transfer of WT or *Gpr37*−/− pMΦ primed with vehicle or NPD1 (30 nM) for 24 h administered 6 h after inoculation of *Gpr37*−/− mice with *L.m.* (1× 10⁷, I.P.). **f** Survival curves of *Gpr37*−/− mice treated with WT or *Gpr37*−/− pMΦ primed with vehicle or NPD1. Sample sizes are indicated in the brackets. **g** Time course of rectal temperature 0 h, 6 h, or 24 h after *L.m.* inoculation. Sample sizes are indicated in the brackets. Data are expressed as the mean ± s.e.m. and statistically analyzed by Mantel–Cox test (**b, f**), One-way ANOVA followed by Tukey's post-hoc test (**d**), or Two-way ANOVA followed by Tukey's (**g**).

Technologies, Inc.; D675-10), ROS/RNS kit (ENZO Life Science; ENZ-51001-200), artesunate (Cayman Chemicals; 11817, 30 mM DMSO stock), artemisinin (Cayman Chemicals; 11816, 30 mM DMSO stock), dihydroartemisinin (Abcam; ab142690, 30 mM DMSO stock), and clodronate liposomes (Liposoma B.V.; C-005). NPD1 was gifted from Resolvyx Pharmaceuticals. All reagents were used in the dose and manner described in the figure legends.

**Plasmids**. The human *GPR37-V5*_pLX304 plasmid was purchased from DNASU. Nano-lantern (cAMP-1.6)/pcDNA3 was a gift from Takeharu Nagai (Addgene plasmid 53594; http://n2t.net/addgene:53594; RRID: Addgene_53594) (26), *pGP-CMV-GCaMP6f* was a gift from Douglas Kim & GENIE Project (Addgene plasmid # 40755; http://n2t.net/addgene:40755; RRID: Addgene_40755), and cytoplasmic *EKAR* (EGFP-mRFP) was a gift from Karel Svoboda (Addgene plasmid # 18680; http://n2t.net/addgene:18680; RRID: Addgene_18680). The colony stocks were directly amplified with LB broth with 100 μg/ml ampicillin (Sigma). Purified plasmid DNA was purified using the Qiagen Midi-prep kit and quantified using a NanoDrop 2000 spectrophotometer.

**Flow cytometry**. Flow cytometry analysis was conducted in accordance with the previous studies[49]. Gating strategies are presented in Supplementary Fig. 4. Prior to staining, cells were blocked in a Fc receptor blocking buffer (1% anti-mouse CD16/CD32, 2.4 G2, 2% FBS, 5% NRS, and 2% NMS in HBSS; BD Bioscience) and stained with a standard panel of antibodies. After staining, all cells were suspended in PBS with EDTA. Flow cytometry events were acquired using a BD FACS Canto II flow cytometer using the BD FACS Diva 8 software (BD Bioscience). The FCS files were analyzed using Cytobank (https://www.cytobank.org/cytobank). Details of the antibodies utilized for FACS are provided in Supplementary Table 2.

**Animals**. C57BL/6J (strain 000664) and *Gpr37* knockout (strain 005806) mice were purchased from the Jackson Laboratory and maintained at a Duke University facility under 12 h dark/light cycles with access to food and water ad libitum. Adult male and female mice (8–10 weeks) were used to produce infection models and for in vitro studies. All animal procedures were conducted in accordance with the National Institutes of Health Guide for the Care and Use of Laboratory Animals and approved by the Institutional Animal Care & Use Committee (IACUC) of Duke University. Behavioral testing and measurement of animal phenotype following infection was rigorously conducted by an experimenter blinded to the genotype and treatment of the animals. Genotyping primers are provided in Supplementary Table 3. Wild type mice were discerned by a 237 base pair band. *Gpr37*−/− mice were discerned by a 400 base pair band. Information regarding the

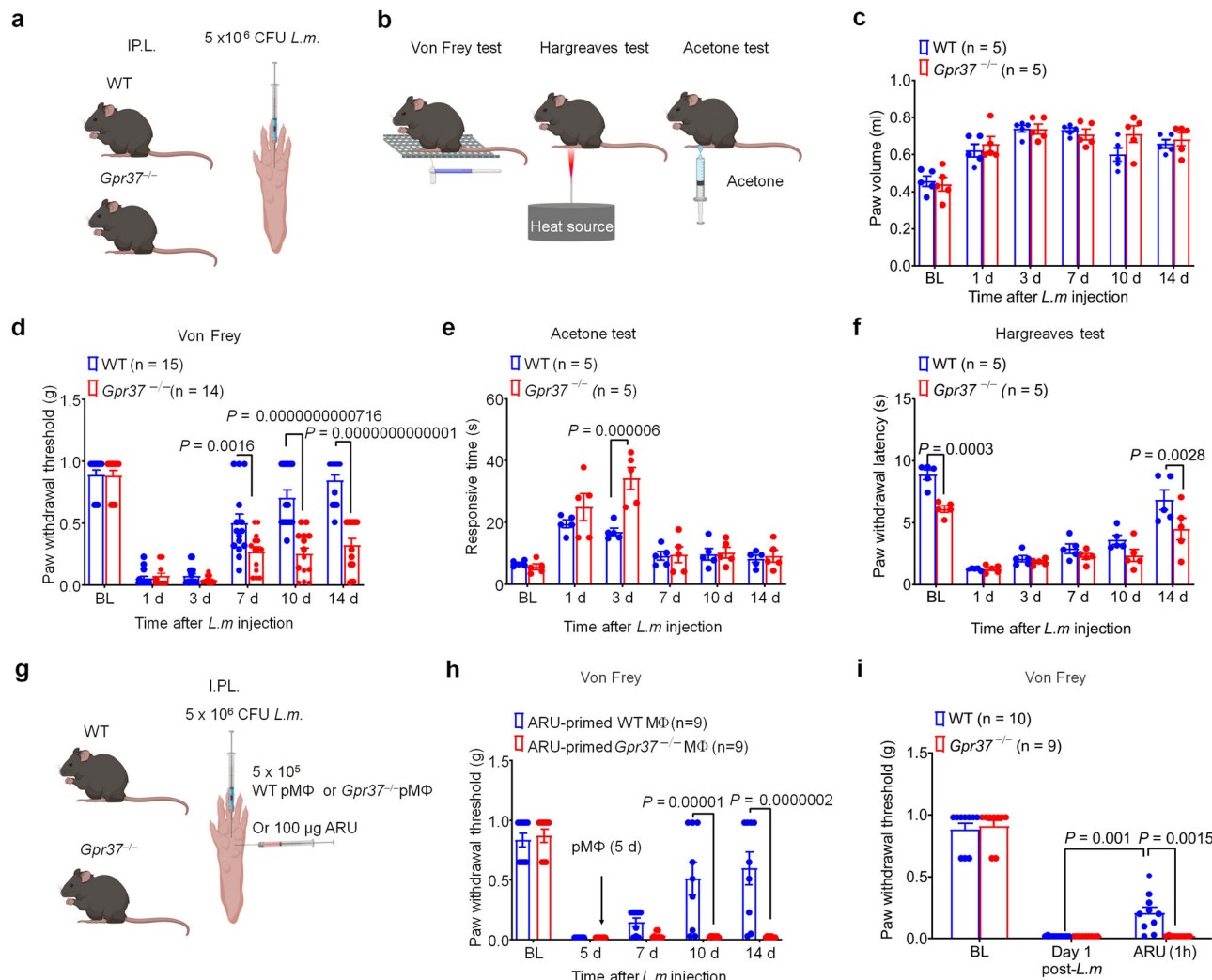

**Fig. 8 GPR37 is also protective against pain caused by bacterial infection. a** Experimental design for *L. m.* infection-induced pain model. *L.m.* ($5 \times 10^6$ CFU) was administered via intraplantar (I.PL.) into the mouse hindpaw, where pain-like behaviors can readily be assessed. **b** Methods used to assess mechanical (Von Frey), heat (Hargreaves), and cold-evoked (acetone) pain-like behaviors in rodents. **c** Time course of hindpaw edema after *L. m* infection. **d** Time course of paw withdrawal thresholds in WT and *Gpr37*$^{-/-}$ mice following *L.m.* inoculation, as determined by Von Frey testing ($n = 14$–15 mice/group). **e** Time course of cold allodynia in WT and *Gpr37*$^{-/-}$ mice following *L.m.* inoculation, as determined by response duration in the plantar acetone test ($n = 5$ mice/group). **f** Time course of heat hyperalgesia in WT and *Gpr37*$^{-/-}$ mice following *L.m.* inoculation, as determined by paw withdrawal latency in the Hargreaves test ($n = 5$ mice/group). **g** Experimental design to test *L.m.* infection-induced pain. *L.m.* ($5 \times 10^6$ CFU) were injected into the hindpaw (intraplantar; I.PL.) of WT or *Gpr37*$^{-/-}$ mice, where pain measurements can readily be assessed. Mice were subsequently treated with ARU-primed WT or *Gpr37*$^{-/-}$ pMΦ or ARU (100 μg) via I.PL. injection and mechanical allodynia was assessed by Von Frey testing. **h** ARU-primed WT or *Gpr37*$^{-/-}$ pMΦ were administered 5 d after *L.m.* inoculation, at which time mice had already developed robust mechanical allodynia, determined by measuring paw withdrawal thresholds. Paw withdrawal thresholds continued to be measured following adoptive transfer at 7 d, 10 d, and 14 d post-inoculation with *L.m.* $n = 9$ mice/group. **i** Paw withdrawal thresholds were measured in WT or *Gpr37*$^{-/-}$ mice at BL, 1 d after *L.m.* inoculation, or 1 d after *L.m.* inoculation 1 h following administration of ARU (100 μg, I.PL.). Data are expressed as the mean ± s.e.m. and were statistically analyzed by two-way ANOVA with Bonferroni's post-hoc test (**c**–**h**).

numbers, genotype, and sex of mice used across all experiments is summarized in Supplementary Table 4.

**Bacteria culture.** *Listeria monocytogenes* was purchased from ATCC (43251). Cells were cultured with Brain Heart Infusion Broth (Millipore Sigma; 53286). Moreover, the number of colonies was counted in Brain Heart Infusion Agar media (Sigma) after overnight culture at 37 °C. CFUs of *L. monocytogenes* were extrapolated from a standard growth curve. The appropriate dilutions were made in sterile phosphate-buffered saline to achieve the desired final bacterial concentration. The actual number of CFU in the inoculum was verified by plating on Brain agar.

**Plasmodium sporozoite culture and isolation.** *Plasmodium berghei* ANKA sporozoites and *Plasmodium falciparum* 3D7-infected human RBC were utilized in

this study. Luciferase-expressing *P. berghei* sporozoites[50] were harvested from freshly dissected salivary glands from *Anopheles stephensi* mosquitoes purchased from the New York University Langone Medical Center Insectary BEI Resources (cat # MRA-134B)[51]. *P. falciparum* 3D7 culture was maintained at 2% parasitemia and 1% hematocrit at 37 °C in a 3% $O_2$, 5% $CO_2$, 92% $N_2$ atmosphere in complete medium (10.44 g/L RPMI 1640 (ThermoFisher Scientific), 25 mM HEPES (ThermoFisher Scientific), pH 7.2, 0.37 mM hypoxanthine (Sigma), 24 mM sodium bicarbonate (Sigma), 0.5% (wt/vol) AlbuMAX II (ThermoFisher Scientific), 25 μg/mL gentamicin (Sigma) supplemented with freshly washed human RBCs (Gulf Coast Regional Blood Center, Houston, TX) every other day[52].

**Mouse models of infection and/or sepsis.** Systemic inflammation was triggered by intraperitoneal (i.p.) injection of LPS (10 mg/kg; Sigma L2630), *Listeria monocytogenes* ($1 \times 10^7$ CFU, a lethal dosage)[53], and by intravenous (i.v.) injection of *Plasmodium berghei* sporozoites ($1 \times 10^4$). Mice were monitored twice daily for

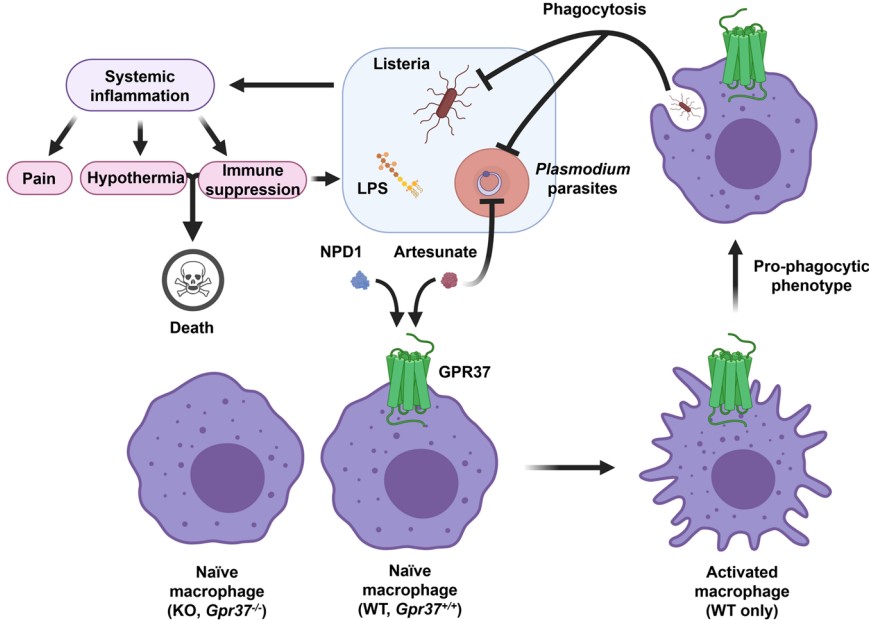

**Fig. 9 Activation of GPR37 in macrophages protects against infection and its sequelae.** Schematic diagram illustrating the mechanism by which GPR37 activation promotes macrophage phagocytosis and clearance of *Listeria* bacteria and *Plasmodium*-infected RBCs.

changes in body weight, body temperature, seizures, and mobility. Cages were cleaned every three days to exclude infection. Mice exhibiting moribund appearance, a body temperature below 30 °C, weight loss exceeding 15%, or severe seizures mice were humanely euthanized in accordance with our approved animal protocol and Duke University guidelines and this was considered the date of death.

**Drug injection**. For i.p. injections, drugs and cells were administered using a sterilized Hamilton micro-syringe (Hamilton) with a 30-gauge needle. For intravenous injections, sporozoites were injected using a 30-gauge needle directly into the tail vein following visualization using a warm water bath. Maximum volume administered for i.p. and i.v. injections were 100 μl. For hindpaw injections, drugs and *L. monocytogenes* were injected (10 μl maximum volume) using a Hamilton micro-syringe with a 30-gauge needle under 2% isoflurane anesthesia with continuous monitoring.

**cAMP BRET assay**. To detect cAMP levels, we transfected a Nano-lantern cAMP plasmid (1 μg) and human *GPR37* plasmid (1 μg) into HEK-293 cells. BRET activity was measured upon addition of 100 μl/well for a final concentration of 10 μM Coelenterazine (Santacruz), 10 μM drug, and 100 μM forskolin (Enzo, # BML-CN100). Bioluminescent activity was measured using an infinite F200 Pro Illuminometer (Tecan Trading AG, Switzerland). The data displayed indicate the values normalized to basal signal levels.

**pERK FRET assay**. Cytoplasmic *EKAR* (1 μg) and human *GPR37-V5* plasmids (1 μg) were transfected in HEK-293 cells. FRET activity was measured using an infinite F200 Pro Illuminometer. The 510/70 and 635/90 barrier filters correspond to the green and red channels. The data displayed indicate the values normalized to basal signal levels.

**cAMP assay**. Mouse peritoneal macrophage cAMP levels were measured using the cAMP-Glo™ Kit (Promega; V1501), performed according to the manufacturer's instructions. Macrophage cells treated with forskolin or artesunate. cAMP-mediated bioluminescence was measured using an infinite F200 Pro Illuminometer (Tecan Trading AG, Switzerland).

**Calcium imaging**. The Human *GPR37* and *Gcamp6F*-transfected Hek293 cells, which lack endogenous voltage-activated calcium and sodium channels, were plated in a cover glass with (in mM) 140 NaCl, 5 KCl, 2 CaCl₂, 2 MgCl₂, 10 HEPES (titrated to pH 7.4 with NaOH). Images of the GCaMP6f positive cells with the excitation wavelength alternating in 488 nm were captured with a cooled ORCA-Flash4.0 Digital CMOS camera (Hamamatsu).

**ELISA**. Mouse ELISA kits for IL-1β, TNF-α, and IL-6, were purchased from R&D (#MLB00C for IL-1β, #DY406 for IL-6, and #DY410 for TNF-α). ELISA was performed using the culture medium, serum, and plasma. Macrophage cells (500,000) were seeded to 24-well plate at 37 °C with RPMI media. Serum was

obtained from mice via facial vein puncture (30–60 μl). The clot was then removed in a refrigerated centrifuge at 2000 × g for 10 min to collect the supernatant (serum). For each ELISA assay, 50 μl of the culture medium or 5 μl of serum were used. The serum cytokine level was measured according to the manufacturer's protocol using a Bio-Rad plate reader.

**RBC isolation and Giemsa staining**. Blood was collected by facial vein puncture (30–60 μl) into a tube containing 10 mM EDTA (Sigma) and 1-unit heparin (Sigma). The plasma was separated by centrifugation (2000 G 10 min). Furthermore, the RBC and WBC were washed with PBS containing 10 mM EDTA for FACS and Giemsa staining. For Giemsa staining, 10 μl of blood were smeared on charged glass slides and fixed with 100% methanol (10 min). After methanol fixation, air-dried slides were stained with 5% Giemsa solution (Sigma) for 20 min. Slides were subsequently washed using distilled water and stored at room temperature until imaging. Images were collected using a ×64 objective on a Zeiss microscope with an AxioCam ICc3 camera affixed. Images were quantified by experimenters blinded to genotype and treatment group.

**Drug docking simulation**. A sequence of human GPR37 was downloaded from the UniProt database (ID: O15354) in *fasta* format. The predicted topology for GPR37 in UniProt was used for seven-transmembrane alignments and long loop identification. Template selection and homology modeling was performed using the automated modeling server GPCR-ModSim[54]. The human M2-Muscarinic Acetylcholine PDB structure (ID: 4MQS) was selected as a template and active conformation of the model was generated. GPCR extracellular loops are essential for receptor activity and second extracellular loop (ECL2) is important for ligand binding. For that reason, we remodeled ECL2 using template structure of human delta opioid (PDB ID: 4N6H) using Prime module of Schrodinger software suite (Schrödinger Release 2018-4: Prime, Schrödinger, LLC, New York, NY, 2018.). All other loops were also refined by Prime module.

The ligands were drawn in the Maestro suite in 2D format and were structurally preprocessed using LigPrep from the Schrodinger Suite (Schrödinger Release 2018-4: LigPrep, Schrödinger, LLC, New York, NY, 2018.). Protonation at a physiological pH (7 ± 2) and energy minimization was performed using the OPLS3 force field. To elucidate the binding mode of all ligands in the binding-site of the homology model of GPR37, docking studies were performed with the help of Autodock4 software[55]. Before the docking, the GPR37 structure was prepared using the AutoDock Tools 4 software.

The stability and intra-molecular conformational changes of the protein, molecular dynamics simulations (MDS) were performed on a 100 ns time scale for the protein–ligand complex. Using the GPU-accelerated DESMOND software[56], the top-scored docking poses were subjected to solvent-explicit, all-atom MDS. The OPLS3 force field was used for model generation of the protein–ligand complex, energy minimization and MDS. The protein was inserted into the POPC lipid bilayer and the full system was immersed in a periodic orthorhombic water box TIP3P. The NPT ensemble class was used with temperature set to 300 K and pressure set to 1.01325 bar.

The structure of Tx14 peptide was modeled using de-novo peptide structure prediction tool PEP-FOLD[57]. The molecular dynamics simulation (MDS) of Tx14 was carried out for 1 μs and 1000 trajectory structures were recorded. Clustering of 1000 trajectory structures into 10 clusters based on atomic RMSDs was performed using the Desmond trajectory clustering tool. The ensemble structure from the largest cluster was chosen for docking input. The structure of GPR37 from the GPR37-NPD1 MDS was used for GPR37-Tx14 docking. To investigate the binding interaction between GRP37 and Tx14, a protein–protein docking was performed for these two protein structures using the ZDOCK Server. Five possible interaction solutions were generated by the ZDOCK server[58]. The docked protein–protein complex was selected based on the maximum number of intermolecular hydrogen bonds established between GPR37 and Tx14 and the stability (RMSD value) of active conformation of GPR37 during MDS.

**Chemical binding assay**. DMSO-soluble chemicals (honokiol, artemisinin, artesunate, or DMSO without drug) were directly loaded on hydrophobic NC membranes (Bio-Rad). Compound-coated membranes were dried and blocked with 1% BSA. Membranes were incubated with lysates obtained from GPR37-expressing HEK293 cell or GFP-expressing cell (mock) lysates for 2 h, followed by detection using an anti-V5 antibody (Cell Signaling, mouse, 1:1000, #80076). Blots were further incubated with an HRP-conjugated secondary antibody (Jackson Immunoresearch, raised in donkey, 1:5000), developed in ECL solution (Pierce), and Bio-Rad ChemiDoc XRS revealed the chemiluminescence signal. The intensity of the dots was analyzed using NIH Image J software.

**HEK293 cell culture and transfection**. The HEK293 Flp-In™ cell line (Invitrogen, R78007) was purchased from the Duke Cell Culture Facility. Cells were cultured in high glucose (4.5 g/L) Dulbecco's Modified Eagle's Medium containing 10% (v/v) fetal bovine serum (Gibco). Transfection (2 μg cDNA/1 × 10^7 cells with 100 μl solution) was performed by electroporation according to the manufacturer's protocol (Lonza, protocol #CM130) when cells reached 70% confluency, and transfected cells were further cultured for 48 h before use.

**Peritoneal macrophage culture**. Peritoneal macrophages (pMΦ) were collected by peritoneal lavage with 10 ml of warm PBS containing 1 mM EDTA. Cells were incubated in RPMI supplemented with 10% FBS at 37 °C for 2 h in a non-coating plate and washed with PBS to eliminate any non-adherent cells. The adherent cells were considered as peritoneal macrophages after 3–5 days (80–90% cells are CD11b and F4/80 positive cells by flow cytometry analysis), by following a standard protocol[59].

**Human macrophage differentiation**. THP-1 cells were purchased from the Duke core facility (ATCC # TIP202). Cells were cultured in high glucose (4.5 g/L) RPMI 1640 Medium containing 10% (v/v) FBS at 37 °C. THP-1 cells were treated in RPMI media containing 250 nM PMA (Sigma) for 24 h and then washed with PBS to eliminate non-adherent cells. The adherent cells were differentiated (mature) human macrophages.

**Phagocytosis assay**. The *L. monocytogenes* or zymosan phagocytosis assay was modified from a previously described protocol[17]. pHrodo® Red Zymosan Bioparticles (diameter, 3 μm, Thermos scientific, Catalog: P35364) and pHrodo® Red-AM-incubated *L. monocytogenes* (diameter, 0.1–0.3 μm) or infected RBC (3 weeks mice after infection, diameter, 5–7 μm) were rinsed and reconstituted in RPMI medium. Particles were centrifuged (1 min at 1000 × g) onto adherent peritoneal macrophages at a ratio of 10:1 to synchronize binding and internalization. After 30 min (for zymosan) and 1 h (for *L. monocytogenes* or iRBC) incubation at 37 °C, non-adherent particles or cells were removed with cold PBS, and cells were fixed with 2% formalin. Four visual fields were photographed under a bright area with epifluorescence microscopy. Quantification of particles ingested by pMΦ was conducted on at least 200 macrophages/condition/well, and triplicates were included for statistical analysis.

**Bacterial colony formation unit assay**. To detect the bacterial clearing activity, the peritoneal lavage macrophage (100 μl at 1 × 10^6 cells/ml) were added to 1 × 10^5 L.m. in 100 μl PBS and incubated for 15–120 min at 37 °C with 100 nM NPD1, 30 μM ARU, or vehicle control with RPMI media (no antibiotics). Cells were repeatedly washing out with PBS for the elimination of extracellular bacteria and lysed with water to detect internal bacterial load. To detect the bacterial colony, serial dilutions were plated on brain heart infusion (BHI) agar plates. The colony formation unit was calculated 24 h later after seeding on the BHI agar plate.

For determination of bacterial burden of the peritoneal fluid, liver, and spleen, mice were sacrificed for 8–48 h after inoculation of 1 × 10^7 L.m. with an application of vehicle or drug. The isolated tissues were homogenized in PBS, and serial dilutions of the homogenate plated on BHI agar plates after measuring of weight, with colony counting 24 h later.

**Measurement of rectal temperature**. The environmental temperature was maintained at 23 ± 0.5 °C and was monitored by a lubricated thermal probe inserted into the rectum of mice (insert depth of ~2 cm). The rectal temperature was measured by the TA-29 flexible Implantable probe (Warner Instruments., Hamden, CT, USA), and TC-344C temperature measuring unit (Warner Instruments., Hamden, CT, USA).

**Pain tests for mechanical, thermal, and cold sensitivity**. Mice were habituated to the environment for at least 2 days before the testing. All the behaviors were tested by an experimenter blinded to genotypes and treatments. Infection-induced pain after administration of 5 × 10^6 CFU *L. m.* (a local inflammation model according to a previous report[4]) was measured in the mouse hindpaws. For testing mechanical sensitivity, mice were confined in boxes (14 × 18 × 12 cm) placed on an elevated metal mesh floor and the ipsilateral hindpaw was stimulated with a series of von Frey hairs with logarithmically increasing stiffness (0.16–2.00 g, Stoelting), presented perpendicularly to the central plantar surface. The 50% paw withdrawal threshold was measured by Dixon's up-down method. Thermal sensitivity was measured using a Hargreaves radiant heat apparatus (IITC Life Science). For the heat-induced withdrawal reflex, the basal paw withdrawal latency was detected around 10–15 s, with a cutoff of 25 s to prevent tissue damage. Acetone (50 μl) was applied through wire mesh flooring onto the plantar surface of the infected hindpaw to produce evaporative cooling, enabling measurement of cold allodynia. The reflex behavior was measured by time of licking, flicking, and guiding time after drug application for 10 min. Paw swelling (edema) after *L. monocytogenes* infection was quantified by the water displacement plethysmometer (Ugo Basile). The plethysmometer is a micro-controlled volume meter specially designed for the accurate measurement of mouse paw swelling. All the behaviors were tested and analyzed blindly.

**Statistical analyses**. All the data were expressed as mean ± s.e.m, as indicated in the figure legends. The sample size for each experiment was based on our previous studies on such examination[17]. All statistical analyses were completed with Prism GraphPad 6.0. Biochemical or behavioral results were analyzed using a two-tailed Student's *t*-test (two groups), One-way ANOVA followed by Tukey's posthoc test (>2 groups); and two-way ANOVA followed by Tukey's posthoc test (>2 groups) or Bonferroni posthoc test (two groups). Survival analyses were performed using a Kaplan–Meier method; statistical significance was analyzed using a log-rank (Mantel–Cox) test. If the overall log-rank test was statistically significant ($P < 0.05$), pairwise comparisons of the survival curves were performed. Because only planned post-hoc comparisons were made specific to the experimental design, corrections for multiple comparisons were not performed; only unadjusted pairwise *P*-values were reported for survival analyses. $P < 0.05$ were considered statistically significant. A summary of all statistical tests is provided in Supplementary Table 5.

**Reporting Summary**. Further information on research design is available in the Nature Research Reporting Summary linked to this article.

## Data availability
All the data are available from the authors upon request. Source data are provided with this paper.

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

## Acknowledgements

We thank Resolvyx Pharmaceuticals for providing NPD1 and Dr. Kuan-Yi Lu at Duke University for providing *P. falciparum*-infected RBCs. Illustrations were created by S.B. and C.R.D. using BioRender. Funding: R.R.J. was supported by Duke University Anesthesiology Research funds. E.R.D. was supported by NIH DP2 AI138239. C.R.D. was supported by the John J. Bonica fellowship from the International Association for the Study of Pain (IASP), NIH T32 GM008600, and Duke University Anesthesiology Research funds. R.R.J. and C.R.D. also received funding from the Duke Microbiome Center.

## Author contributions

S.B., C.R.D., and R-R.J. developed the project. S.B., C.R.D., X.L., X.T., and S.C. conducted experiments. M.T.M. prepared malaria parasites and provided technical advice under the supervision of E.R.D. S.B., C.R.D., X.L., X.T., A.V.B., and Z.W. analyzed the data, C.R.D., S.B., and R-R. J. wrote the paper and M.T.M., E.R.D., S.C., and A.V.B. edited the paper.

## Competing interests

Dr. Ji is a consultant of Boston Scientific and received a research grant from the company. This activity is not related to this study. Drs. Ji and Bang also filed a patent on GPR37-related treatment for inflammation and sepsis. All other authors declare no competing interests.
