## [Peer Review File · Nature Communications]

REVIEWER COMMENTS

Reviewer #1 (Remarks to the Author):

In the current manuscript, Bang et al., demonstrated new functions of GPR37 in controlling infections and sepsis. Of particular interest the anti-malarial drug artesunate stimulates phagocytosis, likely via activating GPR37, providing a new mechanism for the clinically used drug for severe malaria.

I have the following comments. It would be great if the authors can carry out additional experiments to address these points. If that is not feasible at this point, please discuss them in the text.

(1) Do the authors have data on peritoneal leukocyte counts and composition in WT vs GPR37 ^{-/-} mice? It is of interest if NPD1-GPR37 axis affects infiltration of different leukocyte populations.

(2) NPD1 was shown to accelerate resolution of sterile inflammation (Zymosan-induced peritonitis), reducing PMN infiltration and shortening resolution interval (Ri) (Schwab et al., Nature 2007). Is it possible to address if NPD1 also shortens Ri in infectious peritonitis? Along these lines, does Artesunate shares the same pro-resolving action, i.e. shortens Ri?

(3) Does NPD1 protects malaria infection in WT mice? And do NPD1 and Artesunate have additive effect in protecting infection, especially in malaria infection?

(4) Please also include molecular docking of the protein ligand prosaposin-derived peptide (TX14) to GPR37. It is possible that it has different predicted binding sites compared to the small molecule ligands NPD1 and Artesunate.

Reviewer #2 (Remarks to the Author):

The authors recently identified a high macrophage expression of the G-protein coupled receptor GPR37, however the precise role of this receptor in immunity remains unclear. In response to listeria, lipopolysaccharide or malarial infiltration, GPR37-knockout mice experience an increase in death, inflammation indices, and/or hypothermia, compared to wild-type controls. GPR37 agonists include neuroprotection D1 and artesunate, the second being discovered by the authors from a small library of natural compounds during this study. Both agonists are protective in lipopolysaccharide and listeria-induced sepsis, an effect that is lost in GPR37-knockout mice. Mechanistic studies demonstrate that artesunate

binds to GPR37, triggering GPR37-dependent macrophage phagocytosis, supported by macrophage depletion with clodronic acid, and adoptive transfer studies with artesunate-primed macrophages.

This study is interesting, important, well-written, novel, dynamic, biologically plausible, and appropriately controlled. It uses multiple methods for (i) triggering inflammation, (ii) activating and inactivating GPR37, and (iii) assessing inflammation. Across these, the effects appear consistent. These results could contribute to the development of new treatments for sepsis, malaria, listeria and inflammation-associated pain, as well as further investigation into the mechanisms and indications for artesunate and other new and existing GRP37 agonists. It is limited by a relative lack of human work and GRP37-artesunate downstream signalling studies, however further work such as this is discussed briefly in the discussion section. Further exploration of the indices of inflammation resolution would also be interesting. However, on balance, it is a well-performed and well-articulated research study that I believe, aside from a few minor grammatical errors, is ready for publication and of interest to your readership.

Reviewer #3 (Remarks to the Author):

Bang et al have assessed the role of GPR37 in two established models of sepsis and in infection with a mouse model for malaria, *Plasmodium berghei*. They further treat the animals with the natural ligand for the GPR37, NPD1 and model the interaction of GPR37 and a small molecule ARU which acts like the natural ligand and show that treatment with NPD1 or ARU can increase phagocytic activity of macrophages, reduce Il-6 levels and sepsis-induced mortality. They further show through elegant adoptive transfer experiments that this effect can be enhanced by pretreating macrophages with NPD1 or ARU. Overall, this paper is extremely thorough, timely, broadly relevant and well-done. I have one suggestion to improve the manuscript.

In Figure 1f during sepsis induced by Lm survival indicates that some of the effects of GPR37 could be due to a reduced cell intrinsic clearance of Lm. Since Lm can evade macrophage killing and proliferate in professional phagocytes, it would be informative to assess a quantification of bacterial load in the liver or spleen either 3 or 4 days following infection. To my mind that would help distinguish between GRP37-mediated effects on reducing cytokines from sepsis (indicated by the LPS model) and an increased bacterial load which seems to contribute to the effect as the authors observed from the increased mortality in the GPR37 KO mice. The authors performed a similar experiment in Pb using infected erythrocytes as a readout so an analysis of levels of bacterial load would bring the Lm experiments up to the same standard and tease apart relative contributions of bacterial replication and signaling to inflammation.

In addition, in the in vitro experiments the authors list the CFUs of bacteria added to

macrophages but without listing the number of macrophages infected it is hard to compare the conditions they used to other studies thus the multiplicity of infection (MOI) used should be stated for all experiments as well. In this context they should similarly assess bacterial load over a time course of infection through CFU assays of lysed cells to more directly assess the effect of GPR37 on bacterial killing. These two experiments would tie increased phagocytosis to bactericidal activity (indicated in other experiments by NO and LC3 but not directly demonstrated by assessing bacterial load).

Minor points:

Figure 1i – maybe consider an inset or interrupted x axis to zoom in on the difference for Lm infection

Point-by-point response to reviewer comments:

Reviewer #1

Comment 1: [General]: In the current manuscript, Bang et al., demonstrated new functions of GPR37 in controlling infections and sepsis. Of particular interest the anti-malarial drug artesunate stimulates phagocytosis, likely via activating GPR37, providing a new mechanism for the clinically used drug for severe malaria.

I have the following comments. It would be great if the authors can carry out additional experiments to address these points. If that is not feasible at this point, please discuss them in the text.

Response 1: We thank the reviewer for their time and suggestions to improve our manuscript. We have conducted additional experiments and the new results are included in **Fig. 3g-i, Fig. 4e-g, Fig. 5j, Supplementary Fig. 1d-f, Supplementary Fig. 3, and Supplementary Fig. 4f-k**. Please find our responses to each of the questions raised detailed below.

Comment 2: Do the authors have data on peritoneal leukocyte counts and composition in WT vs GPR37 ^{-/-} mice?

Response 2: This is an interesting question and could be reasoned to contribute to the increased lethality observed in GPR37-KO mice. We performed additional FACS experiments to analyze the proportion of leukocytes in peritoneal fluid (PF) in WT vs. GPR37-KO mice under baseline conditions. We did not find significant differences in any of the immune cell types analyzed, which included macrophages (CD11b⁺ F4/80⁺), B-cells (CD19⁺ CD11b⁻), T cells (CD3⁺ CD11b⁻), or neutrophils (Ly6G⁺ F4/80⁻). These data are now provided in **Supplementary Figure 1d-f**.

Comment 3: It is of interest if NPD1-GPR37 axis affects infiltration of different leukocyte populations. NPD1 was shown to accelerate resolution of sterile inflammation (Zymosan-induced peritonitis), reducing PMN infiltration and shortening resolution interval (Ri) (Schwab et al., Nature 2007). Is it possible to address if NPD1 also shortens Ri in infectious peritonitis? Along these lines, does Artesunate shares the same pro-resolving action, i.e. shortens Ri?

Response 3: To address this important question, we analyzed time-dependent PMN infiltration and bacterial load in peritoneal fluid (PF), liver, and spleen after infection with 10⁷ CFU *L.m.*, together with vehicle, NPD1 (500 ng), or ARU (10 mg/kg) treatment (as in Fig. 1), analyzing both factors at 8h, 24h, or 48h post-infection. Interestingly, we observed that both GPR37 agonists could reduce tissue bacterial load, and this effect was abolished in GPR37-KO mice (**Figure 4e-f; Supplementary Fig. 3b-d**). PMN infiltration was significantly reduced by NPD1 or ARU treatment in all three tissues at varying timepoints (**Supplementary Fig. 3e-g**). Using the Gaussian equation, we calculated ψ_{\max} (maximal PMNs), T- ψ_{\max} (time point when PMNs reach ψ_{\max}), ψ_{50} (50% PMN reduction), and T- ψ_{50} (time point corresponding to ~50% PMN reduction), followed by calculation of the resolution interval (interval between T- ψ_{\max} and T- ψ_{50}).

From these data, we found that the resolution interval was not markedly different between vehicle and GPR37 agonist-treated groups (vehicle: 11.07h; NPD1: 18 h; ARU: 16.13 h; $P = 0.961$ vehicle vs. NPD1, $P = 0.9371$ vehicle vs. ARU for P.F; **Supplementary Fig. 3h**). There are several possibilities that we did not see shortening resolution interval (R_i) as reported by Schwab et al. (Nature 2007). First, we used different models in these two studies (*L.m.* bacterial infection vs. zymosan model). Second, there is substantial inhibition of infection by NPD1 and ARU, making it difficult to see the peak response in the treated mice.

Comment 4: Does NPD1 protects malaria infection in WT mice? Do NPD1 and Artesunate have an additive effect in protecting infection, especially in malaria infection?

Response 4: Interestingly, we found that NPD1 treatment (500 ng, I.P.) can reduce the proportion of infected RBCs at 10d post-inoculation with sporozoites (**Supplementary Fig. 4i-k**), but did not significantly alter survival (median survival time WT + vehicle: vs. WT + NPD1+vehicle; **Supplementary Fig. 4f**). Similarly, although the proportion of infected RBCs was reduced by combinatorial treatment with NPD1 and ARU (500 ng NPD1 and 10 mg/kg ARU, administered daily on day 2, 3, and 4), we did not see a significant difference in survival time in NPD1 + ARU treated mice compared to NPD1 + vehicle-treated mice (**Supplementary Fig. 4h**, $P = 0.0602$, although there is a trend toward significance, $n=14$). That being said, we previously found that ARU treatment during the same time window (day 2-4, 10 mg/kg I.P.) was also insufficient to alter the time course of survival in *P.b.*-treated WT mice, which generally exhibit survival ratios of >50%, with the majority of deaths occurring more than 2 weeks after inoculation (**Fig. 5j**). Thus, it is possible that early intervention in WT mice is insufficient to alter survival as the disease burden isn't until later stages, after which time the drug has been eliminated. It is also possible that immune profile change has not reached to the threshold to change survival or our sample size is still insufficient to show mild changes.

Comment 5: Please also include molecular docking of the protein ligand prosaposin-derived peptide (TX14) to GPR37. It is possible that it has different predicted binding sites compared to the small molecule ligands NPD1 and Artesunate.

Response 5: As the reviewer requests, we now include molecular docking of TX14 to GPR37. Interestingly, we were unable to confirm binding of TX14 in the NPD1 binding site of GPR37, prompting us to search for other putative binding sites. We found the second extracellular loop (ECL2) to be a putative binding site, prompting us to perform docking experiments using the ZDOCK server. As shown in **Fig. 3g-i**, in the docked complex of TX14-GPR37, TX14 is positioned at the distal side of ECL2, with GPR37 residues TYR339 and GLN402 interacting with TX14. During the interaction, GPR37 appears to stabilize its RMSD within 1-2 Å. Thus, our data suggests that NPD1 and TX14 may interact with GPR37 via distinct binding sites.

Reviewer #2:

Comment 1: The authors recently identified a high macrophage expression of the G-protein coupled receptor GPR37, however the precise role of this receptor in immunity remains unclear. In response to listeria, lipopolysaccharide or malarial infiltration, GPR37-knockout mice experience an increase in death, inflammation indices, and/or hypothermia, compared to wild-type controls. GPR37 agonists include neuroprotection D1 and artesunate, the second being discovered by the authors from a small library of natural compounds during this study. Both agonists are protective in lipopolysaccharide and listeria-induced sepsis, an effect that is lost in GPR37-knockout mice. Mechanistic studies demonstrate that artesunate binds to GPR37, triggering GPR37-dependent macrophage phagocytosis, supported by macrophage depletion with clodronic acid, and adoptive transfer studies with artesunate-primed macrophages.

This study is interesting, important, well-written, novel, dynamic, biologically plausible, and appropriately controlled. It uses multiple methods for (i) triggering inflammation, (ii) activating and inactivating GPR37, and (iii) assessing inflammation. Across these, the effects appear consistent. These results could contribute to the development of new treatments for sepsis, malaria, listeria and inflammation-associated pain, as well as further investigation into the mechanisms and indications for artesunate and other new and existing GRP37 agonists. It is limited by a relative lack of human work and GRP37-artesunate downstream signalling studies, however further work such as this is discussed briefly in the discussion section. Further exploration of the indices of inflammation resolution would also be interesting. However, on balance, it is a well-performed and well-articulated research study that I believe, aside from a few minor grammatical errors, is ready for publication and of interest to your readership.

Response: We thank the reviewer for their enthusiasm and feedback. We agree that future studies validating these findings in human macrophages and/or clinical samples will be of paramount importance. We did undertake additional experiments to evaluate other indices of inflammation resolution (please see reviewer #1, response #3 and reviewer #3, response #2). The new data indicate that GPR37 agonists attenuate *L.m.*-induced tissue bacterial load and reduce PMN infiltration in peritoneal fluid, liver, and spleen (**Fig. 4e-f, Supplementary Fig. 3**). In addition, we have taken care to correct minor grammatical errors throughout the manuscript.

Reviewer #3:

Comment 1: Bang et al have assessed the role of GPR37 in two established models of sepsis and in infection with a mouse model for malaria, *Plasmodium berghei*. They further treat the animals with the natural ligand for the GPR37, NPD1 and model the interaction of GPR37 and a small molecule ARU which acts like the natural ligand and show that treatment with NPD1 or ARU can increase phagocytic activity of macrophages, reduce IL-6 levels and sepsis-induced mortality. They further show through elegant adoptive transfer experiments that this effect can be enhanced by pretreating macrophages with NPD1 or ARU. Overall, this paper is extremely thorough, timely, broadly relevant and well-done. I have one suggestion to improve the manuscript.

Response 1: We thank the reviewer for their enthusiastic support, their insightful suggestions, and their willingness to provide feedback on our manuscript. Please find our responses to the questions raised in the points below.

Comment 2: In Figure 1f during sepsis induced by Lm survival indicates that some of the effects of GPR37 could be due to a reduced cell intrinsic clearance of Lm. Since Lm can evade macrophage killing and proliferate in professional phagocytes, it would be informative to assess a quantification of bacterial load in the liver or spleen either 3 or 4 days following infection. To my mind that would help distinguish between GPR37-mediated effects on reducing cytokines from sepsis (indicated by the LPS model) and an increased bacterial load which seems to contribute to the effect as the authors observed from the increased mortality in the GPR37 KO mice. The authors performed a similar experiment in Pb using infected erythrocytes as a readout so an analysis of levels of bacterial load would bring the Lm experiments up to the same standard and tease apart relative

Response 2: This is an interesting question, which we have addressed by performing a time-dependent analysis of bacterial load in peritoneal fluid (PF), liver, and spleen after infection with 10^7 CFU *L.m.*, together with vehicle, NPD1 (500 ng), or ARU (10 mg/kg) treatment (as in Fig. 1). As the reviewer suggests, we found that both GPR37 agonists could reduce tissue bacterial load (as determined by CFU assays), and this effect was abolished in GPR37-KO mice (**Fig. 4e-f; Supplementary Fig. 3b-d**). As described in response to comment #3 raised by reviewer #1, we also found that PMN infiltration was significantly reduced by NPD1 or ARU treatment with varying kinetics in different tissues (**Supplementary Fig. 3e-h**). In addition, our previous data suggest GPR37 activation with NPD1 attenuates *L.m.*-induced upregulation of IL-6 (**Fig. 1h**). Thus, we propose that the protective effects of GPR37 are multimodal, involving regulations of PMN infiltration, bacterial clearance, and cytokine storm (e.g., IL-6), effects which are likely interrelated.

Comment 3: In addition, in the in vitro experiments the authors list the CFUs of bacteria added to macrophages but without listing the number of macrophages infected it is hard to compare the conditions they used to other studies thus the multiplicity of infection (MOI) used should be

stated for all experiments as well. In this context they should similarly assess bacterial load over a time course of infection through CFU assays of lysed cells to more directly assess the effect of GPR37 on bacterial killing. These two experiments would tie increased phagocytosis to bactericidal activity (indicated in other experiments by NO and LC3 but not directly demonstrated by assessing bacterial load).

Response 3: We thank the reviewer for this important comment. As requested, we now list the MOI for each experiment in the figure legends where appropriate. As the reviewer suggests, we examined the bacterial clearing activity in lysate from peritoneal macrophages co-cultured with *L.m.* (1×10^5 macrophages; 1×10^4 CFU *L.m.*; MOI = 0.1), which revealed that NPD1 (100 nM) and ARU (30 μ M) treatment potentiated *L.m.* clearance in WT, but not GPR37-KO macrophages (**Fig. 4g**). These experiments also revealed that GPR37-KO macrophages exhibit elevated internal bacterial load, suggesting deficient clearance. Thus, as the reviewer indicated, these data directly tie phagocytic function to bactericidal activity, strengthening our mechanistic data.

Comment 4: Minor points: Figure 1i – maybe consider an inset or interrupted x axis to zoom in on the difference for Lm infection

Response 4: As the reviewer suggests, we now include these data in separate bar graph panels (**Fig. 1i-k**).

REVIEWERS' COMMENTS

Reviewer #1 (Remarks to the Author):

The revision is satisfactory, as the authors have fully addressed all my comments by carrying out additional experiments. The molecular docking results are particularly interesting in that GPR37 appears to have distinct binding sites for NPD1 versus TX-14, suggesting that these two ligands could initiate different intracellular signaling cascades.

I have no further critiques for the authors.

Reviewer #2 (Remarks to the Author):

All issues have been addressed satisfactorily.

Reviewer #3 (Remarks to the Author):

The authors addressed my concerns and I think the manuscript is ready for publication.

REVIEWERS' COMMENTS

Reviewer #1 (Remarks to the Author):

The revision is satisfactory, as the authors have fully addressed all my comments by carrying out additional experiments. The molecular docking results are particularly interesting in that GPR37 appears to have distinct binding sites for NPD1 versus TX-14, suggesting that these two ligands could initiate different intracellular signaling cascades.

I have no further critiques for the authors.

Reviewer #2 (Remarks to the Author):

All issues have been addressed satisfactorily.

Reviewer #3 (Remarks to the Author):

The authors addressed my concerns and I think the manuscript is ready for publication.

We thank the reviewers for their constructive comments. We are very pleased that all the reviewers are satisfied with our responses.